# Generalized Smooth Stochastic Variational Inequalities: Almost Sure Convergence and Convergence Rates

## Abstract

This paper focuses on solving a stochastic variational inequality (SVI) problem under relaxed smoothness assumption for a class of structured non-monotone operators. The SVI problem has attracted significant interest in the machine learning community due to its immediate application to adversarial training and multi-agent reinforcement learning. In many such applications, the resulting operators do not satisfy the smoothness assumption. To address this issue, we focus on a weaker generalized smoothness assumption called $\alpha$-symmetric. Under $p$-quasi sharpness and $\alpha$-symmetric assumptions on the operator, we study clipped projection (gradient descent-ascent) and clipped Korpelevich (extragradient) methods. For these clipped methods, we provide the first almost-sure convergence results without making any assumptions on the boundedness of either the stochastic operator or the stochastic samples. Furthermore, we provide the first in-expectation unbiased convergence rate results for these methods under a relaxed smoothness assumption.

## 1 Introduction

This paper focuses on the stochastic variation inequality (SVI) problem, which consists of finding a point $u^* \in U$, such that

$$\langle F(u^*), u - u^* \rangle \geq 0 \qquad \text{for all } u \in U,$$

where the operator $F(\cdot)$ is specified as the expected value of a stochastic operator $\Phi(\cdot, \cdot) : U \times \Xi \to \mathbb{R}^m$, i.e.,

$$F(u) = \mathbb{E}[\Phi(u, \xi)] \qquad \text{for all } u \in U,$$

where $\xi \in \Xi$ is a random vector. Variational Inequality (VI) problems encompass many practical applications, such as optimization, min-max problems, and multi-agent games. In particular, they play a vital role in modeling equilibrium problems where it's important to capture an interaction between many agents. In machine learning literature, the increasing focus on VIs is due to their relevance to generative adversarial networks (GANs) Gemp & Mahadevan (2018); Gidel et al. (2019), actor-critic methods Pfau & Vinyals (2016), adversarial training, and multi-agent reinforcement learning Sokota et al. (2022); Kotsalis et al. (2022). In many such applications, the corresponding operator is defined as an expected value of stochastic or finite sum of operators, which motivates us to study SVIs. One of the pivotal works Nemirovski (2004); Juditsky et al. (2011) on SVIs proposed and studied the celebrated Mirror-Prox method under assumptions on monotonicity and Lipschitz continuity of an operator.[1] These assumptions become classical for the analysis of first-order methods for solving SVIs Beznosikov et al. (2022); Hsieh et al. (2019; 2020); Loizou et al. (2021).

In adversarial and multi-agent training, where the corresponding operator is a gradient of a highly non-linear neural network model, these classical assumptions might not be satisfied. It is well-known that one possible remedy for such non-convergent behavior is in clipping, normalization, or adaptive stepsizes, such as ADAM (Kingma & Ba, 2015). This effect might be explained by the experiment conducted in Zhang et al. (2020). In this work, authors observed that when training deep neural

---

[1]The well-studied Mirror-Prox method Nemirovski (2004) has been proven to be optimal for solving VIs under strong monotonicity and Lipschitz continuity assumptions. In fact, this method is the stochastic version of the classical extragradient method.

network, the norm of the hessian of the loss function correlates with a norm of a gradient along the optimization trajectory.

This observation motivated Zhang et al. (2020) to introduce a new and more realistic assumption on the linear growth of the hessian. This led to a great number of works in optimization investigating new assumptions on generalized smoothness and convergence behavior of classical gradient (Li et al., 2023), normalized (Chen et al., 2023), clipped (Koloskova et al., 2023), and adaptive methods (Wang et al., 2023; Zhang et al., 2024). Despite this progress in optimization, there are only a few works on generalized smooth min-max (Xian et al., 2024) and VI (Vankov et al., 2024) problems. This motivates us to delve into investigation of first-order methods for generalized smooth SVIs.

## 1.1 RELATED WORK

**Weaker Assumption on SVIs.** More work has focused on stochastic methods for SVI under more relaxed assumptions to develop and analyze the methods applicable to broader problem classes. In particular, some studies have explored SVIs under pseudo-monotonicity Kannan & Shanbhag (2019), quasi-monotonicity Loizou et al. (2021), co-coercivity Beznosikov et al. (2023) and quasi-sharpness Vankov et al. (2023). Diakonikolas et al. (2021) showed that such conditions may not be satisfied even in two played Markov games and introduced the weakest known structured non-monotone assumption. Later, weak Minty SVIs were studied Pethick et al. (2023); Choudhury et al. (2023); Alacaoglu et al. (2024) under Lipschitz's assumption on the operator. In our work, we consider the generalized smooth assumption that goes beyond the existing settings.

**Normalized and Clipped Methods for SVIs.** Jelassi et al. (2022) studied the performance of normalized stochastic gradient descent-ascent and ADAM and suggested the crucial role of normalization for training GANs. It is worth noting that, with the right clipping parameters, clipped and normalized step sizes are equivalent up to a constant. Another line of works Gorbunov et al. (2022) focuses on smooth SVIs under heavy-tail noise. Using Lipschitz's continuity of operator and the right choice of clipping parameters, the authors showed a high probability convergence rate for the clipped stochastic Korpelevich method. Recent work Xian et al. (2024) considered generalized smooth stochastic nonconvex strongly-concave min-max problems and provided $\mathcal{O}(\frac{1}{\sqrt{K}})$ convergence rates for variants of stochastic gradient sescent ascent (SGDA) with normalized stepsizes. Due to the specific structure of the minmax problem and the fact that the gradient of the corresponding function is nonmonotone in one variable and strongly monotone in another, it is difficult to compare this work with ours. Moreover, in this work, the crucial part of the analysis is in the fact that the norm of a gradient can be upper bounded by a function residual. One can not use such bounds in SVIs due to the absence of function values. In our analysis, we develop a new technique to bound the operator norm in almost sure (*a.s.*) sense and in expectation.

**Stochastic Analysis of Clipped Methods for Generalized Smooth Optimization.** In Zhang et al. (2020), authors analyzed clipped gradient method under *a.s.* bounded error assumption. Koloskova et al. (2023) showed that the gradient method with standard clipping may not converge to a solution even with small stepsizes. The authors analyzed clipped gradient descent as a biased method and provided a convergence rate for non-convex functions. Later, Li et al. (2024) developed a new technique allowing to bound stochastic gradients by the function value residual along the optimization trajectory, which helps to find the convergence rate for the gradient with the right choice of stepsizes. In our work, we do not make an assumption on *a.s.* bounded noise and bounded stochastic gradients. Furthermore, we provide not only in-expectations but also *a.s.* convergence of the considered clipped methods.

**Contributions.** In light of the existing literature, we consider stochastic VIs with $p$-quasi sharp *generalized smooth* $\alpha$-symmetric operators. We assume a bounded variance of the noise and do not use a restrictive assumption of bounded stochastic operators or bounded samples. Our key contributions are summarized below (see also Table 1):

- We provide the first known analysis of the clipped stochastic clipped projection method (clipped SGDA) for solving stochastic generalized smooth VIs with $p$-quasi sharp and $\alpha$-symmetric operators. The key feature of our analysis is the use of cleverly chosen clipped *stochastic stepsizes* $\gamma_k$. We use two different samples of stochastic the operator, one for clipping stepsizes $\gamma_k$ and

NEW

another for the direction of the method update. This choice allows us to separate the clipping part from the stochastic error and analyze the method in an unbiased manner. To show *a.s.* convergence, we prove that the series of clipped *stochastic* stepsizes is not summable *a.s.*, i.e. $\mathbb{P}(\sum_{k=0}^{\infty} \mathbb{E}[\gamma_k \mid \mathcal{F}_{k-1}] = \infty) = 1$.

- We also provide convergence rate for stochastic clipped projection method for $\alpha \leq 1/2$. For $p = 2$ we achieve $\mathcal{O}(k^{-1})$ last iterate convergence. For $p > 2$, we show the best iterate convergence rate of $\mathcal{O}(k^{-2(1-q)/p})$, where $1 > q > 1/2$ is a parameter of the stepsize choice.

- We provide the first known analysis of the stochastic clipped Korpelevich method for solving stochastic generalized smooth VIs with $p$-quasi sharp and $\alpha$-symmetric operators. By reusing clipping stepsizes $\gamma_k$ for both iterates updates $h_k$ and $u_k$, we separate stochastic stepsize from the stochastic error, similar to the projection method analysis. To show *a.s.* convergence, we prove that the series of clipped *stochastic* stepsizes is not summable *a.s.*, i.e. $\mathbb{P}(\sum_{k=0}^{\infty} \mathbb{E}[\gamma_k \mid \mathcal{F}_{k-1}] = \infty) = 1$.

- Moreover, we prove in-expectation convergence rates for the stochastic clipped Korpelevich methods for $\alpha \leq 1/2$. For $p = 2$, we show the last iterate sublinear convergence rate $\mathcal{O}(k^{-1})$. For $p > 2$, we show the best iterate convergence rate of $\mathcal{O}(k^{-2(1-q)/p})$, where $1 > q > 1/2$ is a parameter of the stepsize choice.

- Finally, we present numerical experiments where we compare the performance of the methods with proposed stochastic clipping for different stepsize parameter $q > 1/2$ and quasi-sharpness parameter $p$.

|  | Stochastic Projection | Stochastic Korpelevich |
|---|---|---|
| $p > 0$ | Asym (Thm 3.2) | Asym (Thm 4.2) |
| $p = 2$ | $\mathcal{O}\left(\dfrac{D_0}{k^2} + \dfrac{\sigma^2(C_F + \sigma)^2}{\mu^2 k}\right)$ | $\mathcal{O}\left(\dfrac{(C_F + \sigma)^2 K^2 D_0}{\mu^2 k^2} + \dfrac{(\sigma^2 + K_1 \sigma^{2\alpha})(C_F + \sigma)^2}{\mu^2 k}\right)$ |
| $p > 2$ | $\mathcal{O}\left(\dfrac{(D_0 + \sigma^2)^{2/p}(C_F + \sigma)^{2/p}}{\mu^{2/p} k^{2(1-q)/p}}\right)$ | $\mathcal{O}\left(\dfrac{\sigma^{4/p}(D_0 + \sigma^2 + K_1 \sigma^{2\alpha})^{2/p}(C_F + \sigma)^{2/p}}{\mu^{2/p} k^{2(1-q)/p}}\right)$ |

Table 1: Summary of convergence rate results showing the decrease of certain performance measures with the number $k$ of iterations. We use "Asym" as an abbreviation for asymptotic almost sure convergence results. For $p$-quasi sharp operators, with $p = 2$, and for stochastic projection and Korpelevich methods, the performance measures are $D_k = \mathbb{E}[\text{dist}^2(u_k, U^*)]$ and $D_k = \mathbb{E}[\text{dist}^2(h_k, U^*)]$, respectively. For $p > 2$, the performance measure for both methods is $D_k^{\text{best}} = \min_{t=0,\ldots,k} D_t$. The constant $C_F$ denotes the upper bound on $\mathbb{E}[\|F(u_k)\|]$ and $\mathbb{E}[\|F(h_k)\|]$ for stochastic projection and Korpelevich methods, respectively.

The rest of the paper is organized as follows. In Section 2, we define the assumption on the operator class we consider and define the first-order methods we focus on. In Section 3 we show the almost sure convergence result of clipped stochastic projection method. In Section 4, we provide *a.s.* convergence results and in-expectation convergence rates for the clipped stochastic Korpelevich method. In Section 5, we conduct experiments on solving generalized smooth SVIs and compare the performance of the stochastic clipped projection and Korpelevich method for different problem and stepsize parameters. Section 6 concludes our work and presents some further research directions.

## 2 PRELIMINARIES

In this section, we provide the necessary concepts and assumptions for the considered SVI problem. We start with a standard definition; the operator $F$ is said to be Lipschitz continuous on a set $U$ if there exists $L > 0$ such that

$$\|F(u) - F(v)\| \leq L\|u - v\| \quad \text{for all } u, v \in U. \tag{1}$$

So far, Lipschitz continuity of operator was the most common assumption to study SVIs Nemirovski (2004); Yousefian et al. (2014; 2017); Hsieh et al. (2019); Loizou et al. (2021); Alacaoglu et al. (2024). However, this assumption does not hold in modern deep-learning applications. Based on the experiments provided in Zhang et al. (2020), the norm of Jacobian of the operator correlates with the

norm of the operator. Recent work Chen et al. (2023) proposed a new, more realistic, and weaker assumption termed $\alpha$-symmetric given as follows: $F$ is $(L_0, L_1)$-smooth operator on a set $U$ if

$$\|F(u) - F(v)\| \leq \left(L_0 + L_1 \max_{\theta \in (0,1)} \|F(w_\theta)\|^\alpha\right) \|u - v\| \quad \text{for all } u, v \in U, \tag{2}$$

where $w_\theta = \theta u + (1 - \theta)v$ and $\alpha \in (0, 1]$. When the operator $F(\cdot)$ is $L$-Lipshitz continuous, it satisfies (2) with $L_0 = L$ and $L_1 = 0$. The class of $\alpha$-symmetric operators includes the class of $(L_0, L_1)$-smooth operators and coincides with it when an operator is differentiable for $\alpha = 1$. Given that the class of $\alpha$-symmetric operators includes the class of $(L_0, L_1)$-smooth and Lipschitz continuous operators we focus on this class in our work.

**Assumption 2.1.** Given a convex set $U \subseteq \mathbb{R}^m$, the operator $F(\cdot) : U \to \mathbb{R}^m$ is $\alpha$-symmetric over $U$, i.e., for some $\alpha \in (0, 1]$ and $L_0, L_1 \geq 0$, we have for all $u, v \in U$,

$$\|F(u) - F(v)\| \leq \left(L_0 + L_1 \max_{\theta \in (0,1)} \|F(w_\theta)\|^\alpha\right) \|u - v\|, \tag{3}$$

where $w_\theta = \theta u + (1 - \theta)v$.

An alternative characterization of $\alpha$-symmetric operators has been proved in Chen et al. (2023), as given in the following proposition.

**Proposition 2.2** (Chen et al. (2023), Proposition 1). *Let $U \subseteq \mathbb{R}^m$ be a nonempty convex set and let $F(\cdot) : U \to \mathbb{R}^m$ be an operator. Then, the following statements hold:*

(a) *$F(\cdot)$ is $\alpha$-symmetric with $\alpha \in (0, 1)$ and constants $L_0, L_1 \geq 0$ if and only if the following relation holds for all $y, y' \in U$,*

$$\|F(y) - F(y')\| \leq \|y - y'\|(K_0 + K_1\|F(y')\|^\alpha + K_2\|y - y'\|^{\alpha/(1-\alpha)}), \tag{4}$$

*where $K_0 = L_0(2^{\alpha^2/(1-\alpha)} + 1)$, $K_1 = L_1 2^{\alpha^2/(1-\alpha)} 3^\alpha$, and $K_2 = L_1^{1/(1-\alpha)} 2^{\alpha^2/(1-\alpha)} 3^\alpha (1-\alpha)^{\alpha/(1-\alpha)}$.*

(b) *$F(\cdot)$ is $\alpha$-symmetric with $\alpha = 1$ and constants $L_0, L_1 \geq 0$ if and only if the following relation holds for all $y, y' \in U$,*

$$\|F(y) - F(y')\| \leq \|y - y'\|(L_0 + L_1\|F(y')\|)\exp(L_1\|y - y'\|). \tag{5}$$

Proposition 2.2 is useful for our analysis, since it describes an $\alpha$-symmetric operator by using two points $y, y' \in U$, and bypasses the evaluation of $\max_{\theta \in (0,1)} \|F(w_\theta)\|^\alpha$. The solution set for the variational inequality problem defined by the set $U$ and operator $F$, denoted $U^*$, is given by

$$U^* = \{u^* \in U \mid \langle F(u^*), u - u^* \rangle \geq 0 \text{ for all } u \in U\}.$$

Throughout this paper, we make the following assumption on the set $U$ and the solution set.

**Assumption 2.3.** The set $U \subseteq \mathbb{R}^m$ is a nonempty closed convex set, and the solution set $U^*$ is nonempty and closed.

We use $p$-quasi sharp operators since, for such operators, the inner product between an operator value $F(u)$ and $u - u^*$ is positive, which is crucial in our analysis. Moreover, this class encompasses both strongly monotone and strongly coherent operators and aligns with the class of operators that satisfy the Saddle-Point Metric Subregularity Wei et al. (2021) for $p > 2$. The formal definition of the $p$-quasi sharpness property is presented below.

**Assumption 2.4.** The operator $F(\cdot) : U \to \mathbb{R}^m$ has a $p$-quasi sharpness property over $U$ relative to the solution set $U^*$, i.e., for some $p > 0$, $\mu > 0$, and for all $u \in U$ and $u^* \in U^*$,

$$\langle F(u), u - u^* \rangle \geq \mu \operatorname{dist}^p(u, U^*). \tag{6}$$

For solving the SVI problem, we consider stochastic variants of projection and Korpelevich Korpelevich (1976) methods, where stochastic approximations $\Phi(u_k, \xi_k)$ and $\Phi(h_k, \xi_k^1)$ are used, respectively, instead of the directions $F(u_k)$ and $F(h_k)$. The iterates of each of the stochastic methods are defined as follows:

Stochastic projection method:

$$u_{k+1} = P_U(u_k - \gamma_k \Phi(u_k, \xi_k)), \tag{7}$$

Stochastic Korpelevich method:

$$\begin{aligned} u_k &= P_U(h_k - \gamma_k \Phi(h_k, \xi_k^1)), \\ h_{k+1} &= P_U(h_k - \gamma_k \Phi(u_k, \xi_k^2)), \end{aligned} \tag{8}$$

where $\{\gamma_k > 0\}$ is a sequence of stochastic stepsizes, and $u_0, h_0 \in U$ are arbitrary deterministic initial points[2]. At each operator evaluation of these stochastic methods, a random sample $\xi_k$ is drawn according to the distribution of the random variable $\xi$. We assume that the stochastic approximation error $\Phi(u, \xi) - F(u)$ is unbiased and has finite variance, leading to the following formal assumption.

**Assumption 2.5.** The random sample $\xi$ is such that for all $u \in U$,

$$\mathbb{E}[\Phi(u, \xi) - F(u) \mid u] = 0, \qquad \mathbb{E}[\|\Phi(u, \xi) - F(u)\|^2 \mid u] \leq \sigma^2.$$

Our proof techniques in the following sections can be applied to analyze the (*a.s.*) convergence and convergence rate of the stochastic Popov (Popov, 1980) method with an appropriate selection of stochastic clipping. However, due to space constraints, we leave this exploration for future research.

## 3 PROJECTION METHOD

Common approaches to developing convergent methods for generalized smooth optimization and VI problems are normalized or clipping stepsizes. We focus on the latter one and present stepsizes for the stochastic projection method for $\alpha$-symmetric operators:

$$\gamma_k = \beta_k \min\left\{1, \frac{1}{\|\Phi(u_k, \xi_k^2)\|}\right\}, \tag{9}$$

where $\beta_k > 0$ for all $k \geq 0$ and $\xi_k^2$ is a random variable, such that $\xi_k^2$ and $\xi_k$ are independent conditionally on $u_k$. In other words, at every iteration of the projection method, having $u_k$, two independent samples of the stochastic operator are drawn: (1) $\Phi(u_k, \xi_k)$ for the direction of update and (2) $\Phi(u_k, \xi_k^2)$ for clipping stepsize $\gamma_k$. We define the sigma-algebra $\mathcal{F}_k$ for the method:

$$\mathcal{F}_k = \{\xi_0, \xi_0^2, \ldots, \xi_k, \xi_k^2\} \qquad \text{for all } k \geq 0, \tag{10}$$

with $\mathcal{F}_{-1} = \emptyset$. In the sequel, we provide important results on the behavior of the iterates of the clipped stochastic projection method.

### 3.1 ALMOST SURE CONVERGENCE

The following lemma establishes a key relation for the iterate sequence $\{u_k\}$ generated by the stochastic projection method with stochastic clipping stepsizes. Its proof is in Appendix B.1

**Lemma 3.1.** *Let Assumptions 2.1, 2.3, 2.4, 2.5 hold, and $\{u_k\}$ be the iterate sequence generated by stochastic projection method (7) with stepsizes $\gamma_k$ defined in (9). Let parameter $\beta_k$ be such that $\sum_{k=0}^{\infty} \beta_k = \infty$ and $\sum_{k=0}^{\infty} \beta_k^2 < \infty$. Then, the following relation holds almost surely for all $k \geq 0$,*

$$\mathbb{E}[\|u_{k+1} - u^*\|^2 \mid \mathcal{F}_{k-1}] \leq \|u_k - u^*\|^2 - 2\mu\mathbb{E}[\gamma_k \mid \mathcal{F}_{k-1}]\text{dist}^p(u_k, U^*) + 3\beta_k^2(2\sigma^2 + 1). \tag{11}$$

*Furthermore, almost surely, we have*

$$\sum_{k=0}^{\infty} \mathbb{E}[\gamma_k \mid \mathcal{F}_{k-1}]\,\text{dist}^p(u_k, U^*) < \infty, \tag{12}$$

*and, the sequence $\{\|u_k - u^*\|^2\}$ is bounded almost surely for all $u^* \in U^*$.*

---

[2]The results easily extend to the case when the initial points are random as long as $\mathbb{E}[\|u_0\|^2]$ and $\mathbb{E}[\|h_0\|^2]$ are finite.

In the conventional analysis of methods for SVIs with Lipschitz continuous operators, the sequence $\{\gamma_k\}$ of stepsizes is deterministic and such that $\sum_k \gamma_k = \infty$. In our case, $\gamma_k$ is a random variable, and to show *a.s.* convergence we have to show that the sequence $\{\mathbb{E}[\gamma_k \mid \mathcal{F}_{k-1}]\}$ is not summable. We do so, providing a sequence of lower bounds for series $\sum_{k=0}^{\infty} \mathbb{E}[\gamma_k \mid \mathcal{F}_{k-1}]$ and by showing that random variable $\|F(u_k)\|$ is *a.s.* upper bounded for all $k \geq 0$. In the next theorem, we present the first results on *a.s.* convergence of the stochastic projection method.

**Theorem 3.2.** *Let Assumptions 2.1, 2.3, 2.4, and 2.5 hold, and $\{u_k\}$ be the iterate sequence generated by stochastic projection method (7) with stepsizes $\gamma_k$ defined in (9). Let parameter $\beta_k$ be such that $\sum_{k=0}^{\infty} \beta_k = \infty$ and $\sum_{k=0}^{\infty} \beta_k^2 < \infty$. Then, the iterates $u_k$ converge almost surely to a point $\bar{u}$ such that $\bar{u} \in U^*$ almost surely.*

The full proof of Theorem 3.2 can be found in Appendix B.2. Notice that in an unconstrained setting ($U = \mathbb{R}^m$) according to Theorems 3.1 and 3.2 in Koloskova et al. (2023) for any clipping parameters $\beta > 0, c > 0$ there exist a stochastic gradient operator $\nabla_\xi f(\cdot)$ which satisfies Assumptions 2.1, 2.4 (with $p = 2$), 2.5 for which there exists a fixed point $\hat{v}$ of a standard clipping with one-sample which there exists a solution

NEW

$$\mathbb{E}_\xi[\beta_k \min\{1, \frac{c}{\|\nabla_\xi f(\hat{v})\|}\}] = \hat{v} \quad \text{and} \quad \|\mathbb{E}_\xi[\nabla_\xi f(\hat{v})]\| \geq \sigma^2/12,$$

where $C > 0$ is a constant independent from a step sizes parameter $\beta_k$. This observation leads to an unavoidable bias in one-sample clipped SGD (Koloskova et al., 2023). In contrast, by using two sample in clipped projection method we overcome this problem and provide *a.s.* convergence to a solution.

### 3.2 CONVERGENCE RATE

The difficulty of the convergence rate analysis is in the randomness of stepsizes $\gamma_k$. To show in-expectation convergence, we can take a total expectation on both sides of equation (11) of Lemma 3.1. However, since $\gamma_k$ is a random variable, we have to provide a lower bound on $\mathbb{E}[\gamma_k \text{dist}^p(u_k, U^*)]$. With this goal in mind, in the next lemma we show that the sequence $\{\mathbb{E}[\|F(u_k)\|]\}$ of expected norms is bounded. The proof of the lemma is in Appendix B.3.

**Lemma 3.3.** *Let Assumption 2.1 hold, with $\alpha \in (0, 1/2]$, Assumptions 2.3, 2.4, 2.5 hold, and $\{u_k\}$ be iterate sequence generated by stochastic projection method (7) with stepsizes $\gamma_k$ defined in (9). Let parameter $\beta_k$ be such that $\sum_{k=0}^{\infty} \beta_k = \infty$, and $\sum_{k=0}^{\infty} \beta_k^2 < \infty$. Then, the sequence $\{\mathbb{E}[\|F(u_k)\|]\}$ is bounded by some constant $C_F > 0$.*

To prove the preceding lemma, we show that the expected norms of the operator are bounded by some constant $C_F$ on the trajectory of the method. To show this, we use the properties of the method and the generalized smoothness of the operator in Proposition 2.2 to obtain that for all $k \geq 0$, and arbitrary solution $v^*$,

$$\|F(u_k)\| \leq \|F(u_k) - F(v^*)\| + \|F(v^*)\|$$
$$\leq \|u_k - v^*\|(K_0 + K_1\|F(v^*)\|^\alpha + K_2\|u_k - v^*\|^{\alpha/(1-\alpha)}) + \|F(v^*)\|. \quad (13)$$

Notice that by taking an expectation in (13), the RHS is undefined for $\alpha > 1/2$. For $\alpha \in (0, 1/2]$, using (13) and boundedness of $\mathbb{E}[\|u_k - v^*\|]$ we achieve the desired bound on $\mathbb{E}[\|F(u_k)\|]$. Using this result, in the next theorem, we provide a convergence rate for the projection method with clipping.

**Theorem 3.4.** *Let Assumption 2.1, with $\alpha \in (0, 1/2]$, and Assumptions 2.3, 2.4, 2.5 hold. Let $\{u_k\}$ be the sequence generated by stochastic projection method (7) with stepsizes $\gamma_k$ defined in (9). Let $D_k = \mathbb{E}[\text{dist}^2(u_k, U^*)]$ and $C_F$ be an upperbound on $\mathbb{E}[\|F(u_k)\|]$. Then, we have:*

***Case** $p = 2$. Let $\beta_k = \frac{2}{a(2+k)}$ with $a = \mu \min\left\{1, \frac{1}{2(C_F+\sigma)}\right\}$. Then, the following inequality holds*

$$D_{k+1} \leq \frac{8D_0}{k^2} + \frac{6(2\sigma^2 + 1)}{a^2 k} \qquad \text{for all } k \geq 1. \quad (14)$$

***Case** $p \geq 2$. Let $\beta_k = \frac{b}{(k+1)^q}$, where $1/2 < q < 1$ and $b > 0$. Then, the following inequality holds*

$$\bar{D}_k \leq \frac{(1-q)^{2/p}\left(D_0 + 3b^2(2\sigma^2 + 1)/(2q-1)\right)^{2/p}}{(ab)^{2/p}\left((k+1)^{1-q} - 2^{1-q}\right)^{2/p}} \qquad \text{for all } k \geq 1, \quad (15)$$

*where $\bar{D}_k = \mathbb{E}[\text{dist}^2(\bar{u}_k, U^*)]$, $\bar{u}_k = (\sum_{t=0}^k \beta_t)^{-1} \sum_{t=0}^k \beta_t u_t$, and $a = \mu \min\left\{1, \frac{1}{2(C_F+\sigma)}\right\}$.*

For the simplicity of convergence rate comparison, assume $2(C_F + \sigma) \geq 1$. Then, from Theorem 3.4 we obtain $\mathcal{O}(\frac{D_0}{k^2} + \frac{\sigma^2(C_F+\sigma)^2}{\mu^2 k})$ last iterate convergence rate for $p = 2$, and $\mathcal{O}(\frac{(D_0+\sigma^2)^{2/p}(C_F+\sigma)^{2/p}}{\mu^{2/p}k^{2(1-q)/p}})$ average (or best) iterate convergence rate for $p > 2$ with $q \in (1/2, 1)$. It is worth mentioning that obtained rates are unbiased, unlike the analysis in Koloskova et al. (2023). However, it comes with the price of two oracle calls per iteration. For $p = 2$, the rate from Theorem 3.4 matches the rate $\mathcal{O}(\frac{1}{k})$ obtained in Theorem 4.3 (Loizou et al., 2021) for SGDA under stronger assumption on quasi-strong monotonicity and Lipschitz continuity of the operator. The rate for $p > 2$ is new in the stochastic case and generalizes the convergence results in deterministic setting (Vankov et al., 2024). The proof of Theorem 3.4 is in Appendix B.4.

# 4 KORPELEVICH METHOD

The stepsizes for the stochastic Korpelevich method for $\alpha$-symmetric operators are as given below

$$\gamma_k = \beta_k \min\left\{1, \frac{1}{\|\Phi(h_k, \xi_k^1)\|}\right\}, \tag{16}$$

where $\beta_k > 0$ for all $k \geq 0$ and $\xi_k^1$ is a random variable associated with the stochastic approximation $\Phi(h_k, \xi_k^1)$ of $F(h_k)$. We define the sigma-algebra $\mathcal{F}_k$ for the method, as follows:

$$\mathcal{F}_k = \{\xi_0^1, \xi_0^2, \ldots, \xi_k^1, \xi_k^2\} \qquad \text{for all } k \geq 0, \tag{17}$$

with $\mathcal{F}_{-1} = \emptyset$. Notice that to obtain $h_{k+1}$ from a point $u_k$, the stepsize $\gamma_k$ clips $\Phi(h_k, \xi_k^1)$, not the stochastic approximation $\Phi(u_k, \xi_k^2)$ of the operator at point $u_k$, i.e., we have

$$h_{k+1} = P_U\left(u_k - \beta_k \min\left\{1, \frac{1}{\|\Phi(h_k, \xi_k^1)\|}\right\} \Phi(u_k, \xi_k^2)\right). \tag{18}$$

Thus, sample $\xi_k^2$ is drawn after $\xi_k^1$, and $\Phi(h_k, \xi_k^1)$ is measurable with respect to $\mathcal{F}_{k-1} \cup \xi_k^1$. This property of the stochastic Korpelevich method with clipping stepsizes $\gamma_k$ is crucial for further convergence analysis of the method. In the sequel, we provide important results on the behavior of the iterates of the clipped stochastic Korpelevich method.

## 4.1 ALMOST SURE CONVERGENCE

In the forthcoming lemma, we provide some basic relations that hold almost surely for the iterates of the stochastic Korpelevich method with clipped stochastic stepsize.

**Lemma 4.1.** *Let Assumptions 2.1, 2.3, 2.4, and 2.5 hold. Also, let $\{h_k\}$ and $\{u_k\}$ be iterates generated by stochastic Korpelevich method (8) with stepsizes $\gamma_k$ defined in (16) and with parameter $\beta_k$ such that $\sum_{k=0}^\infty \beta_k = \infty$ and $\sum_{k=0}^\infty \beta_k^2 < \infty$. Then, the following relation holds almost surely*

$$\mathbb{E}[\|h_{k+1} - u^*\|^2 \mid \mathcal{F}_{k-1}] \leq \|h_k - u^*\|^2 - \frac{1}{2}\|h_k - u_k\|^2 - 2\mu\mathbb{E}[\gamma_k \mid \mathcal{F}_{k-1}]\text{dist}^p(u_k, U^*)$$
$$+ 6\beta_k^2(\sigma^2 + C_e(\beta_k, \alpha)\sigma^{2\alpha}) \qquad \textit{for all } k \geq 0, \tag{19}$$

*where $C_e(\beta_k, \alpha) = K_1$, when $\alpha \in (0, 1)$, and $C_e(\beta_k, \alpha) = \exp(L_1\beta_k)$, when $\alpha = 1$. Moreover, the following relations hold almost surely,*

$$\sum_{k=0}^\infty \mathbb{E}[\gamma_k \mid \mathcal{F}_{k-1}]\text{dist}^p(u_k, U^*) < \infty, \quad \sum_{k=0}^\infty \|h_k - u_k\|^2 < \infty. \tag{20}$$

*Furthermore, the sequence $\{\|h_k - u^*\|\}$ is bounded almost surely for all $u^* \in U^*$.*

The proof of Lemma 4.1 is in Appendix C.1.

In a standard analysis of the Korpelevich method for SVI with Lipschitz operators Kannan & Shanbhag (2019); Vankov et al. (2023), *a.s.* convergence results were achieved for a deterministic

sequence $\{\gamma_k\}$. In our case, similarly to projection method analysis, $\{\gamma_k\}$ is a sequence of random variables, which makes the analysis of the methods more difficult and involved. By the choice of stepsizes $\gamma_k$ as given in (16), the following relation holds true

$$\mathbb{E}[\gamma_k \langle \Phi(u_k, \xi_k^2) - F(u_k), u_k - u^* \rangle \mid \mathcal{F}_{k-1}] = 0 \qquad \text{for all } k \geq 0. \tag{21}$$

To prove (21) for the stochastic Korpelevich method, we note that the clipping stepsize is using $\|\Phi(h_k, \xi_k^1)\|$, which decouples from $\Phi(u_k, \xi_k^2)$ by properly using conditional expectation. Specifically, we first take the expectation conditioned on $\mathcal{F}_{k-1} \cup \xi_k^1$. Since $\gamma_k$ is measurable with respect to $\mathcal{F}_{k-1} \cup \xi_k^1$, we use Assumption 2.5 and the law of the total expectation. Interestingly, we do not have to take another sample for the clipping in the stochastic Korpelevich method, as we have done in the stochastic projection method. Thus, to perform one iteration, we use two oracle calls in both methods.

Using Lemma 4.1, we next present the almost sure convergence of the clipped Korpelevich method.

**Theorem 4.2.** *Let Assumptions 2.1, 2.3, 2.4, and 2.5 hold and $\{h_k\}$, $\{u_k\}$ be iterates generated by stochastic Korpelevich method (8) with stepsizes $\gamma_k$ defined in (16). Let parameter $\beta_k$ be such that $\sum_{k=0}^{\infty} \beta_k = \infty$, and $\sum_{k=0}^{\infty} \beta_k^2 < \infty$. Then, the iterates $h_k$ and $u_k$ converge almost surely to a point $\bar{u}$ such that $\bar{u} \in U^*$ almost surely.*

To prove *a.s.* convergence, we firstly show that $\sum_{k=0}^{\infty} \mathbb{E}[\gamma_k \mid \mathcal{F}_{k-1}] = \infty$ *a.s.*, by providing a sequence of lower bounds on $\mathbb{E}[\gamma_k \mid \mathcal{F}_{k-1}]$, using *a.s.* boundedness of $\|h_k - u^*\|$ of Lemma 4.1, and proving that $\|F(h_k)\|$ is *a.s.* bounded. The full proof can be found in Appendix C.2.

## 4.2 CONVERGENCE RATE

We start our analysis by taking the total expectation on both sides of equation (19) from Lemma 4.1. For further analysis, similar to the clipped stochastic projection methods, the challenge lies in the randomness of the stepsizes $\gamma_k$. To handle this, firstly, we establish a lower bound for $\mathbb{E}[\gamma_k \text{dist}^p(u_k, U^*)]$ by showing that the sequence $\{\mathbb{E}[\|F(u_k)\|]\}$ of expected norms remains bounded, as shown in the next lemma. The proof of the lemma can be found in Appendix B.3.

**Lemma 4.3.** *Let Assumption 2.1, with $\alpha \in (0, 1/2]$, and Assumptions 2.3, 2.4, 2.5 hold. Let $\{u_k\}$, $\{h_k\}$ be iterates generated by stochastic Korpelevich method (8) with stepsizes $\gamma_k$ defined in (16) and the parameter $\beta_k$ such that $\sum_{k=0}^{\infty} \beta_k = \infty$ and $\sum_{k=0}^{\infty} \beta_k^2 < \infty$. Then, $\mathbb{E}[\|F(h_k)\|]$ is bounded by some constant $C_F > 0$ for all $k \geq 0$.*

Similarly to the analysis presented in Section 3, we bound $F(h_k)$ by using a triangle inequality and the property of $\alpha$-symmetric operators, and by taking the total expectation, we obtain

$$\mathbb{E}[\|F(u_k)\|] \leq K_0 \mathbb{E}[\|u_k - v^*\|] + K_2 \mathbb{E}[\|u_k - v^*\|^{\alpha/(1-\alpha)}] + \|F(v^*)\| + K_1 \|F(v^*)\|^\alpha. \tag{22}$$

We can show that the preceding bound has a finite expectation only for $0 < \alpha \leq 1/2$, which motivates the restriction on $\alpha$ in Lemma 4.3. Equipped with the boundedness of the sequence $\{\mathbb{E}[\|F(h_k)\|]\}$ of expected norms of the operator along the iterates $\{h_k\}$, we present the next convergence rate theorem.

**Theorem 4.4.** *Let Assumption 2.1, with $\alpha \in (0, 1/2]$, and Assumptions 2.3, 2.4, 2.5 hold. Let $\{u_k\}$, $\{h_k\}$ be iterates generated by stochastic Korpelevich method (8) with stepsizes $\gamma_k$ defined in (16). Let $D_k = \mathbb{E}[\text{dist}^2(h_k, U^*)]$ and $C_F$ be an upperbound on $\mathbb{E}[\|F(h_k)\|]$ then the following results holds:*

**Case** $p = 2$. *Let* $\beta_k = \dfrac{2}{a\left(\frac{2d}{a} + k\right)}$, *with* $a = \mu \min\left\{1, \frac{1}{2(C_F + \sigma)}\right\}$, $d = \max\{4\mu, 2\sqrt{3}(K_0 + K_1 + K_2)\}$ *where $K_0, K_1$, and $K_2$ are from Proposition 2.2(a). Then, the following relation holds*

$$D_{k+1} \leq \frac{8d^2 D_0}{a^2 k^2} + \frac{12(\sigma^2 + K_1 \sigma^{2\alpha})}{a^2 k} \qquad \text{for all } k \geq 1. \tag{23}$$

**Case** $p \geq 2$. *Let $\beta_k = \frac{b}{(k+1)^q}$, where $1/2 < q < 1$ and $0 < b \leq \min\left\{\frac{1}{4\mu}, \frac{1}{2\sqrt{3}(K_0 + K_1 + K_2)}\right\}$. Then, the following inequality holds for all $k \geq 1$,*

$$\bar{D}_k \leq \frac{2^{2(p-2)/p}(1-q)^{2/p}\left(D_0 + 6b^2(\sigma^2 + K_1 \sigma^{2\alpha})(2\sigma^2 + 1)/(2q-1)\right)^{2/p}}{(ab)^{2/p}\left((k+1)^{1-q} - 2^{1-q}\right)^{2/p}}, \tag{24}$$

*where $\bar{D}_k = \mathbb{E}[\text{dist}^2(\bar{u}_k, U^*)]$, $\bar{u}_k = (\sum_{t=0}^{k} \beta_t)^{-1} \sum_{t=0}^{k} \beta_t u_t$, and $a = \mu \min\left\{1, \frac{1}{2(C_F + \sigma)}\right\}$.*

The proof of Theorem 4.4 is provided in Appendix C.4. For the simplicity of convergence rate comparison, assume $2(C_F+\sigma) \geq 1$ and $K_0+K_1+K_2 \geq \frac{2\mu}{\sqrt{3}}$. Then by denoting $K = K_0+K_2+K_3$, from Theorem 4.4, we obtain $\mathcal{O}(\frac{(C_F+\sigma)^2 K^2 D_0}{\mu^2 k^2} + \frac{(\sigma^2+K_1\sigma^{2\alpha})(C_F+\sigma)^2}{\mu^2 k})$ last iterate convergence for $p = 2$, and $\mathcal{O}\left(\frac{\sigma^{4/p}(D_0+\sigma^2+K_1\sigma^{2\alpha})^{2/p}(C_F+\sigma)^{2/p}}{\mu^{2/p}k^{2(1-q)/p}}\right)$ average (or best) iterate convergence rate for $p > 2$ with $q \in (1/2, 1)$. In both cases $p = 2$ and $p > 2$ the convergence rate of clipped stochastic projection method in Theorem 3.4 and the rate of clipped stochastic Korpelevich method in Theorem 4.4 have the same dependency in $k$. For $p = 2$ the rate from Theorem 4.4 matches the rate $\mathcal{O}(\frac{1}{k})$ obtained in Proposition 5 (Kannan & Shanbhag, 2019) for stochastic Korpelevich method under stronger assumption on strong pseudo monotonicity and Lipschitz continuity of the operator. For $p > 2$, the obtained rate is new in stochastic case and generalize the results in deterministic setting for Lipschitz continuous operators (Wei et al., 2021) and $\alpha$-symmetric operators (Vankov et al., 2024). **NEW**

## 5 NUMERICAL EXPERIMENTS

We study the performance of the clipped stochastic projection and Korpelevich methods, for different values of parameters $\alpha > 0$ and $p > 0$. Despite the absence of analysis, we also implement the clipped stochastic Popov method with $\gamma_k = \beta_k \min\{1, \frac{1}{\|F(h_k)\|}, \frac{1}{(\|u_k-h_{k-1}\|+1)^{\alpha/(1-\alpha)}}\}$: **NEW**

$$u_{k+1} = P_U(u_k - \gamma_k\Phi(h_k,\xi_k)), \quad h_{k+1} = P_U(u_{k+1} - \gamma_{k+1}\Phi(h_k,\xi_k)),$$

where $u_0, h_0 \in U$ are arbitrary deterministic initial. We consider an unconstrained SVI($\mathbb{R}^2, F$) with the following stochastic operator

$$\Phi(u,\xi) = \begin{bmatrix} \text{sign}(u_1)|u_1|^{p-1} + u_2 \\ \text{sign}(u_2)|u_2|^{p-1} - u_1 \end{bmatrix} + \xi,$$

where $\xi$ is a random vector with independent zero-mean Gaussian entries and with variance $\sigma^2 = 1$. Then, $F(u) = \mathbb{E}[\Phi(u,\xi)]$ is an $\alpha$-symmetric and $p$-quasi sharp operator due to Vankov et al. (2024). We set these parameters to be $\{(\alpha \approx 0.33, p = 2.5), (\alpha \approx 0.5, p = 3.0), (\alpha \approx 0.8, p = 6.0)\}$. We also compare our results with the projection method that uses the same sample clipping, meaning stepsizes $\gamma_k$ clip $\|\Phi(u_k,\xi_k)\|$ instead of a different sample $\|\Phi(u_k,\xi_k^2)\|$.

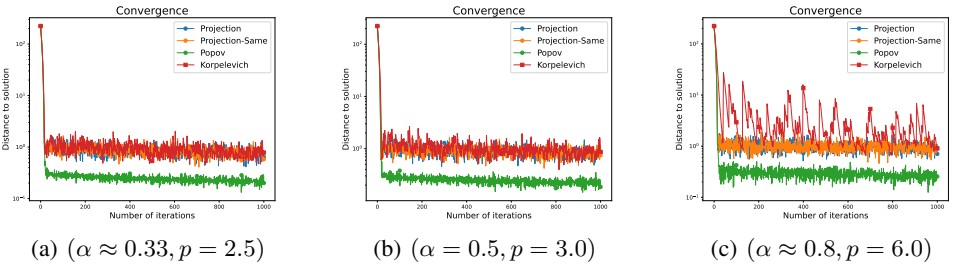

(a) $(\alpha \approx 0.33, p = 2.5)$  (b) $(\alpha = 0.5, p = 3.0)$  (c) $(\alpha \approx 0.8, p = 6.0)$

Figure 1: Comparison of the clipped stochastic projection, same-sample projection, Korpelevich, and Popov methods with $\beta = 100/(100 + k^{1/2+\epsilon})$.

In Figure 1, we plot an average distance over twenty runs to the solution set as a function of the number of iterations. In particular, the stepsizes for clipped stochastic projection and Korpelevich methods are chosen according to Theorems 3.4 and 4.4, respectively, with $\beta_k = \frac{100}{100+k^q}$ for $q = 1/2 + \epsilon$ with $\epsilon > 0$. Note that, according to Theorems 3.4 and 4.4, the parameter $q$ should be greater than $1/2$; meanwhile, the rates in these theorems are better for smaller choices of $q$. We also set $\beta_k = \frac{100}{100+k^q}$ for clipped stochastic Popov method and the clipped stochastic projection method using the same sample $\Phi(u_k,\xi_k)$ for clipping.

Based on this experiment, we made three important observations. Firstly, the clipped stochastic projection method and same-sample clipped stochastic projection method show similar results despite the fact that the same-sample stochastic projection method has a biased error. Secondly, for $\alpha \leq 1/2$ as predicted in theory (Theorems 3.4, 4.4), both projection and Korpelevich methods show

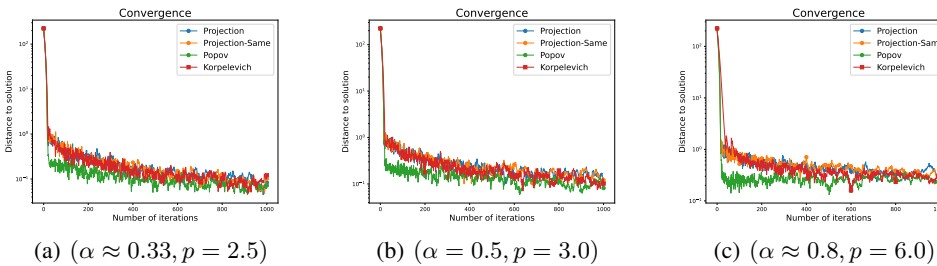

(a) ($\alpha \approx 0.33, p = 2.5$)    (b) ($\alpha = 0.5, p = 3.0$)    (c) ($\alpha \approx 0.8, p = 6.0$)

Figure 2: Comparison of the clipped stochastic projection, same-sample projection, Korpelevich, and Popov methods with $\beta = 100/(100 + k^{1-\epsilon})$.

in-expectation convergence, while for $\alpha > 1/2$ Korpelevich method has less stable performance. Finally, despite the fact that for stochastic Lipschitz SVI, Korpelevich method outperforms the projection method, we don't see this in generalized smooth SVIs.

Next, we investigate the performance of the methods for larger values of $q$. In Figure 2, we set $q = 1 - \epsilon$, and corresponding $\beta_k = \frac{100}{100 + k^{1-\epsilon}}$ and run all four methods for the same problem parameter setting. We observe that for all considered $\alpha$, despite the theory, a larger choice of $q$  FIX improved the performance of all methods in the $\sigma$-neighborhood. Furthermore, it seems that larger values of $q$ help to stabilize the clipped stochastic Korpelevich method for $\alpha > 1/2$.

Additionally, we conducted an experiment studying the robustness of the methods with a larger choice of the initial parameter value $\beta_0$, which are included in Appendix D.

## 6 CONCLUSION

This paper studied the SVI problem under generalized smooth and structured non-monotone assumptions. Specifically, we consider $\alpha$-symmetric and $p$-quasi-sharp operators, a class of generalized smooth and structured non-monotone operators for SVIs. For this wide class of operators, we proved the first-known almost sure convergence of clipped stochastic projection and Korpelevich methods for all parameters $p$. We also provided $\mathcal{O}(1/k)$ convergence rate for both considered methods when the operator is $p$-quasi sharp with $p = 2$. For $p > 2$ we provided $\mathcal{O}(k^{-2(1-q)/p})$ average (or best) iterate convergence rate for both methods, where $q$ is a stepsizes parameter $1/2 < q < 1$. Despite the generality of our results, there are still open questions remain. In particular, it would be interesting to know if it is possible to show in-expectation convergence rates for $\alpha$-smooth SVI $\alpha > 1/2$. Another attractive direction of further research in generalized smooth SVIs is in relaxation of $p$-quasi sharpness assumption to Minty ($\mu = 0$) or weak Minty conditions. We also believe that our technique for proving almost sure convergence and in-expectation rates can be used for the analysis of other methods whose stepsizes are random variables, for example, clipped stochastic Popov method or first-order methods with adaptive stepsizes.

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

## A    TECHNICAL LEMMAS

In our analysis, we use the properties of the projection operator $P_U(\cdot)$ given in the following lemma.

**Lemma A.1.** *(Theorem 1.5.5 and Lemma 12.1.13 in Facchinei & Pang (2003)) Given a nonempty convex closed set $U \subset \mathbb{R}^m$, the projection operator $P_U(\cdot)$ has the following properties:*

$$\langle v - P_U(v), u - P_U(v) \rangle \leq 0 \quad \text{for all } u \in U, v \in \mathbb{R}^m, \tag{25}$$

$$\|u - P_U(v)\|^2 \leq \|u - v\|^2 - \|v - P_U(v)\|^2 \quad \text{for all } u \in U, v \in \mathbb{R}^m, \tag{26}$$

$$\|P_U(u) - P_U(v)\| \leq \|u - v\| \quad \text{for all } u, v \in \mathbb{R}^m. \tag{27}$$

In the forthcoming analysis, we use Lemma 11 Polyak (1987), which is stated below.

**Lemma A.2.** *[Lemma 11 Polyak (1987)] Let $\{v_k\}, \{z_k\}, \{a_k\}$, and $\{b_k\}$ be nonnegative random scalar sequences such that almost surely for all $k \geq 0$,*

$$\mathbb{E}[v_{k+1} \mid \mathcal{F}_k] \leq (1 + a_k)v_k - z_k + b_k, \tag{28}$$

*where $\mathcal{F}_k = \{v_0, \ldots, v_k, z_0, \ldots, z_k, a_0, \ldots, a_k, b_0, \ldots, b_k\}$, and a.s. $\sum_{k=0}^{\infty} a_k < \infty$, $\sum_{k=0}^{\infty} b_k < \infty$. Then, almost surely, $\lim_{k \to \infty} v_k = v$ for some nonnegative random variable $v$ and $\sum_{k=0}^{\infty} z_k < \infty$.*

As a direct consequence of Lemma A.2, when the sequences $\{v_k\}, \{z_k\}, \{a_k\}, \{b_k\}$ are deterministic, we obtain the following result.

**Lemma A.3.** *Let $\{\bar{v}_k\}, \{\bar{z}_k\}, \{\bar{a}_k\}, \{\bar{b}_k\}$ be nonnegative scalar sequences such that for all $k \geq 0$,*

$$\bar{v}_{k+1} \leq (1 + \bar{a}_k)\bar{v}_k - \bar{z}_k + \bar{b}_k, \tag{29}$$

*where $\sum_{k=0}^{\infty} \bar{a}_k < \infty$ and $\sum_{k=0}^{\infty} \bar{b}_k < \infty$. Then, $\lim_{k \to \infty} \bar{v}_k = \bar{v}$ for some scalar $\bar{v} \geq 0$ and $\sum_{k=0}^{\infty} \bar{z}_k < \infty$.*

**Lemma A.4.** *Let $X$ be a non-negative random variable such that $\mathbb{E}[X^\rho]$ is defined for some $\rho \geq 1$, and $\mathbb{E}[X^\rho] \neq 0$, then for every $a > 0$ it holds*

$$\mathbb{P}(X > a(\mathbb{E}[X^\rho])^{1/\rho}) \leq \frac{1}{a^\rho}. \tag{30}$$

*Proof.* Let $Y = X^\rho$. By the conditions of the lemma, the expectation $\mathbb{E}[Y] = \mathbb{E}[X^\rho]$ is well defined. Then, by Markov's inequality:

$$\mathbb{P}(X > a(\mathbb{E}[X^\rho])^{1/\rho}) = \mathbb{P}(Y > a^\rho \mathbb{E}[X^\rho])$$
$$\leq \frac{\mathbb{E}[X^\rho]}{a^\rho \mathbb{E}[X^\rho]}.$$

∎

**Lemma A.5.** *Let $a_1, a_2$ be nonnegative scalar and $p > 0$. Then the following inequality holds:*

$$(a_1 + a_2)^p \leq 2^{p-1}(a_1^p + a_2^p).$$

*Proof.* Let $a = (a_1, a_2), b = (1, 1)$, then by Hölder inequality:

$$a_1 + a_2 = \|ab\|$$
$$\leq \|a\|_p \|b\|_{p/(p-1)}$$
$$\leq (a_1^p + a_2^p)^{1/p}(1 + 1)^{(p-1)/p}.$$

Raising the inequality in the power $p$ we get the desired relation.    ∎

## A.1 AUXILIARY RESULTS

In our analysis we make use of Lemma 3 and Lemma 7 from Stich (2019), as well as the sequences provided in the proofs in Stich (2019).

**Lemma A.6.** *Let $\{r_k\}$ and $\{s_k\}$ be nonnegative scalar sequences that satisfy the following relation*

$$r_{k+1} \leq (1 - a\alpha_k)r_k - b\alpha_k s_k + c\gamma_k^2 \qquad \text{for all } k \geq 0,$$

*where $a > 0$, $b > 0$, $c \geq 0$, and*

$$\gamma_k = \frac{2}{a\left(\frac{2d}{a} + k\right)} \qquad \text{for all } k \geq 0,$$

*where $d \geq a$. Then, for any given $K \geq 0$, the following relation holds:*

$$\frac{b}{W_K} \sum_{k=0}^{K} w_k s_k + ar_{K+1} \leq \frac{8d^2}{aK^2} r_0 + \frac{2c}{aK},$$

*where $w_k = 2d/a + k$, $0 \leq k \leq K$, and $W_K = \sum_{k=0}^{K} w_k$.*

**Lemma A.7.** *For $1 > q \geq 1/2$ and $K \geq 1$, we have*

$$\sum_{t=0}^{K} \frac{1}{(t+1)^q} \geq \frac{1}{1-q}((K+1)^{1-q} - 2^{1-q}). \tag{31}$$

*For $q = 1/2$ and $K \geq 1$,*

$$\sum_{t=0}^{K} \frac{1}{(t+1)^{2q}} \leq \log(K+1). \tag{32}$$

*For $q > 1/2$ and $K \geq 1$,*

$$\sum_{t=0}^{K} \frac{1}{(t+1)^{2q}} \leq \frac{1}{2q-1}. \tag{33}$$

*Proof.* Let $1 > q \geq 1/2$ and $K \geq 1$. Then, it holds

$$\sum_{t=0}^{K} \frac{1}{(t+1)^q} \geq \int_{s=1}^{K} \frac{ds}{(s+1)^q} = \frac{1}{1-q}((K+1)^{1-q} - 2^{1-q}). \tag{34}$$

When $q = 1/2$ and $K \geq 1$, then

$$\sum_{t=0}^{K} \frac{1}{t+1} \leq \int_{s=0}^{K} \frac{ds}{s+1} = \log(K+1). \tag{35}$$

When $q > 1/2$ and $K \geq 1$, we have that

$$\sum_{t=0}^{K} \frac{1}{(t+1)^{2q}} \leq \int_{s=0}^{K} \frac{ds}{(s+1)^{2q}} = \frac{1}{2q-1} - \frac{1}{(2q-1)(K+1)^{2q-1}} < \frac{1}{2q-1}. \tag{36}$$

$\blacksquare$

# B PROJECTION METHOD ANALYSIS

## B.1 PROOF OF LEMMA 3.1

*Proof.* Let $k \geq 0$ be arbitrary but fixed. From the definition of $u_{k+1}$ in (7), we have $\|u_{k+1} - y\|^2 = \|P_U(u_k - \gamma_k \Phi(u_k, \xi_k)) - y\|^2$ for all $y \in U$. Using the non-expansiveness property of projection operator (27) we obtain for all $y \in U$ and $k \geq 0$,

$$
\begin{aligned}
\|u_{k+1} - y\|^2 &\leq \|u_k - \gamma_k \Phi(u_k, \xi_k) - y\|^2 \\
&= \|u_k - y\|^2 - 2\gamma_k \langle \Phi(u_k, \xi_k), u_k - y \rangle + \gamma_k^2 \|\Phi(u_k, \xi_k)\|^2 \\
&= \|u_k - y\|^2 + \gamma_k^2 \|\Phi(u_k, \xi_k)\|^2 \\
&\quad - 2\gamma_k \langle F(u_k), u_k - y \rangle + 2\gamma_k \langle e_k, u_k - y \rangle,
\end{aligned}
\tag{37}
$$

where $e_k = F(u_k) - \Phi(u_k, \xi_k)$. By the definition of the stepsizes (9), $\gamma_k = \beta_k \min\{1, \frac{1}{\|\Phi(u_k, \xi_k^2)\|}\}$, then the term $\gamma_k^2 \|\Phi(u_k, \xi_k)\|^2$ can be upper bounded as follows

$$
\begin{aligned}
\gamma_k^2 \|\Phi(u_k, \xi_k)\|^2 &= \gamma_k^2 \|\Phi(u_k, \xi_k) - F(u_k) + F(u_k) - \Phi(u_k, \xi_k^2) + \Phi(u_k, \xi_k^2)\|^2 \\
&\leq \beta_k^2 \min\left\{1, \frac{1}{\|\Phi(u_k, \xi_k^2)\|^2}\right\} 3(\|e_k\|^2 + \|e_k^2\|^2 + \|\Phi(u_k, \xi_k^2)\|^2) \\
&\leq 3\beta_k^2 \|e_k\|^2 + 3\beta_k^2 \|e_k^2\|^2 + 3\beta_k^2,
\end{aligned}
\tag{38}
$$

where $e_k = F(u_k) - \Phi(u_k, \xi_k)$, $e_k^2 = F(u_k) - \Phi(u_k, \xi_k^2)$. Thus,

$$
\begin{aligned}
\|u_{k+1} - y\|^2 &\leq \|u_k - y\|^2 - 2\gamma_k \langle F(u_k), u_k - y \rangle \\
&\quad + 2\gamma_k \langle e_k, u_k - y \rangle + 3\beta_k^2 (\|e_k\|^2 + \|e_k^2\|^2 + 1).
\end{aligned}
\tag{39}
$$

Plugging in $y = u^* \in U^*$, where $u^*$ is an arbitrary solution, and using $p$-quasi sharpness we get:

$$
\begin{aligned}
\|u_{k+1} - u^*\|^2 &\leq \|u_k - u^*\|^2 - 2\mu\gamma_k \mathrm{dist}^p(u_k, U^*) \\
&\quad + 2\gamma_k \langle e_k, u_k - u^* \rangle + 3\beta_k^2 (\|e_k\|^2 + \|e_k^2\|^2 + 1).
\end{aligned}
\tag{40}
$$

Using stochastic properties of $\xi_k$ and $\xi_k^2$ imposed by Assumption 2.5, and the conditional independence of $\xi_k$ and $\xi_k^2$, we have:

$$
\mathbb{E}[\gamma_k \langle e_k, u_k - u^* \rangle | \mathcal{F}_{k-1}] = \mathbb{E}[\gamma_k \mid \mathcal{F}_{k-1}] \langle \mathbb{E}[e_k \mid \mathcal{F}_{k-1}], u_k - u^* \rangle = 0.
$$

$$
\mathbb{E}[\|e_k\|^2 \mid \mathcal{F}_{k-1}] \leq \sigma^2, \quad \mathbb{E}[\|e_k^2\|^2 \mid \mathcal{F}_{k-1}] \leq \sigma^2.
$$

Thus, by taking the conditional expectation on $\mathcal{F}_{k-1} = \{\xi_0, \xi_0^2, \ldots, \xi_{k-1}, \xi_{k-1}^2\}$ in relation (40) we obtain for all $u^* \in U^*$ and for all $k \geq 0$:

$$
\begin{aligned}
\mathbb{E}[\|u_{k+1} - u^*\|^2 | \mathcal{F}_{k-1}] &\leq \|u_k - u^*\|^2 + 3\beta_k^2 (2\sigma^2 + 1) \\
&\quad - 2\mu\beta_k \mathbb{E}\left[\min\left\{1, \frac{1}{\|\Phi(u_k, \xi_k^2)\|}\right\} \mid \mathcal{F}_{k-1}\right] \mathrm{dist}^p(u_k, U^*).
\end{aligned}
\tag{41}
$$

The equation (41) satisfies the condition of Lemma A.2 with

$$
v_k = \|u_k - u^*\|^2, \quad a_k = 0, \quad z_k = 2\mu \mathbb{E}[\gamma_k \mid \mathcal{F}_{k-1}] \mathrm{dist}^p(u_k, U^*), \quad b_k = 3\beta_k^2 (2\sigma^2 + 1).
\tag{42}
$$

By Lemma A.2, it follows that the sequence $\{v_k\}$ converges *a.s.* to a non-negative scalar for any $u^* \in U^*$, and almost surely we have

$$
\sum_{k=0}^{\infty} \mathbb{E}[\gamma_k \mid \mathcal{F}_{k-1}] \mathrm{dist}^p(u_k, U^*) < \infty.
\tag{43}
$$

Since the sequence $\{\|u_k - u^*\|^2\}$ converges *a.s.* for all $u^* \in U^*$, it follows that the sequence $\{\|u_k - u^*\|\}$ is bounded *a.s.* for all $u^* \in U^*$. ∎

### B.2 PROOF OF THEOREM 3.2

*Proof.* To show almost sure convergence, we need to show that $\sum_{k=0}^{\infty} \mathbb{E}[\gamma_k \mid \mathcal{F}_{k-1}]$ is not summable almost surely. To do so, we provide a sequence of lower bounds on $\mathbb{E}[\gamma_k \mid \mathcal{F}_{k-1}]$. First, consider the following event:

$$A_k = \{\|e_k^2\| \le 2\mathbb{E}[\|e_k^2\| \mid \mathcal{F}_{k-1}]\},$$

where $e_k^2 = F(u_k) - \Phi(u_k, \xi_k^2)$ is a stochastic error from the sample for the clipping stepsize $\gamma_k$. Then, by the law of total expectation

$$\mathbb{E}[\gamma_k \mid \mathcal{F}_{k-1}] = \mathbb{E}[\gamma_k \mid \mathcal{F}_{k-1} \cup A_k]\mathbb{P}(A_k \mid \mathcal{F}_{k-1}) + \mathbb{E}[\gamma_k \mid \mathcal{F}_{k-1} \cup \overline{A}_k]\mathbb{P}(\overline{A}_k \mid \mathcal{F}_{k-1}). \quad (44)$$

We want to provide a lower bound on $\mathbb{P}(A_k \mid \mathcal{F}_{k-1})$. To do so, we upperbound $\mathbb{P}(\overline{A}_k \mid \mathcal{F}_{k-1})$ using Markov's inequality and Assumption 2.5:

$$\mathbb{P}(\overline{A}_k \mid \mathcal{F}_{k-1}) = \mathbb{P}(\|e_k\| > 2\mathbb{E}[\|e_k^2\| \mid \mathcal{F}_{k-1}]\}) \le \frac{\mathbb{E}[\|e_k^2\| \mid \mathcal{F}_{k-1}]}{2\mathbb{E}[\|e_k^2\| \mid \mathcal{F}_{k-1}])} = \frac{1}{2}. \quad (45)$$

Thus,

$$\mathbb{E}[\gamma_k \mid \mathcal{F}_{k-1}] = \mathbb{E}[\gamma_k \mid \mathcal{F}_{k-1} \cup A_k](1 - \mathbb{P}(\overline{A}_k \mid \mathcal{F}_{k-1})) + \mathbb{E}[\gamma_k \mid \mathcal{F}_{k-1} \cup \overline{A}_k]\mathbb{P}(\overline{A}_k \mid \mathcal{F}_{k-1})$$

$$\ge \frac{1}{2}\mathbb{E}[\gamma_k \mid \mathcal{F}_{k-1} \cup A_k] + \mathbb{E}[\gamma_k \mid \mathcal{F}_{k-1} \cup \overline{A}_k]\mathbb{P}(\overline{A}_k \mid \mathcal{F}_{k-1})$$

$$\ge \frac{1}{2}\mathbb{E}[\gamma_k \mid \mathcal{F}_{k-1} \cup A_k]. \quad (46)$$

By definition of $\gamma_k$, triangle inequality and definition of event $A_k$, it holds

$$\mathbb{E}[\gamma_k \mid \mathcal{F}_{k-1} \cup A_k] = \beta_k \mathbb{E}[\min\left\{1, \frac{1}{\|\Phi(u_k, \xi_k^2)\|}\right\} \mid \mathcal{F}_{k-1} \cup A_k]$$

$$\ge \beta_k \mathbb{E}\left[\min\left\{1, \frac{1}{\|F(u_k)\| + \|e_k^2\|}\right\} \mid \mathcal{F}_{k-1} \cup A_k\right]. \quad (47)$$

Next, we use the definition of the event $A_k$ to provide the next lower bound:

$$\mathbb{E}[\gamma_k \mid \mathcal{F}_{k-1} \cup A_k] \ge \beta_k \mathbb{E}[\min\left\{1, \frac{1}{\|F(u_k)\| + 2\mathbb{E}[\|e_k^2\| \mid \mathcal{F}_{k-1}]}\right\} \mid \mathcal{F}_{k-1} \cup A_k]$$

$$\ge \beta_k \mathbb{E}[\min\left\{1, \frac{1}{\|F(u_k)\| + 2\sigma}\right\} \mid \mathcal{F}_{k-1} \cup A_k]$$

$$= \beta_k \min\left\{1, \frac{1}{\|F(u_k)\| + 2\sigma}\right\}. \quad (48)$$

The first inequality in the preceding equation holds by definition of event $A_k = \{\|e_k^2\| \le 2\mathbb{E}[\|e_k^2\| \mid \mathcal{F}_{k-1}]\}$, the second inequality holds dues to Assumption 2.5 on the noise and Jensen inequality, $\mathbb{E}[\|e_k^2\| \mid \mathcal{F}_{k-1}] \le \sqrt{\mathbb{E}[\|e_k^2\|^2 \mid \mathcal{F}_{k-1}]} \le \sigma$, and the last equality holds since $\|F(u_k)\|$ is measurable in $\mathcal{F}_{k-1}$. Hence, it follows that

$$\sum_{k=0}^{\infty} \beta_k \min\left\{1, \frac{1}{\|F(u_k)\| + \sigma}\right\} \le \sum_{k=0}^{\infty} \mathbb{E}[\gamma_k \mid \mathcal{F}_{k-1}]. \quad (49)$$

Now, we want to show that $\sum_{k=0}^{\infty} \beta_k \min\left\{1, \frac{1}{\|F(u_k)\|+\sigma}\right\}$ is not summable almost surely, i.e. $\mathbb{P}(\sum_{k=0}^{\infty} \beta_k \min\left\{1, \frac{1}{\|F(u_k)\|+\sigma}\right\} = \infty) = 1$. We will do so by showing *a.s.* boundedness of $\|F(u_k)\|$ for all $k \ge 0$, using property of $\alpha$-symmetric operators. To estimate $\|F(u_k)\|$, we add and subtract $F(v^*)$, where $v^* \in U^*$ is an arbitrary but fixed solution, and get

$$\|F(u_k)\| = \|F(u_k) - F(v^*) + F(v^*)\| \le \|F(u_k) - F(v^*)\| + \|F(v^*)\|.$$

Define the following event:

$$A = \{\omega \in \Omega : \exists C(\omega) \in \mathbb{R} \text{ s.t.} \|u_k(\omega) - v^*\| \le C(\omega) \, \forall \, k \ge 0\}.$$

Based on results of Lemma 3.1, the sequence $\{\|u_k - u^*\|\}$ is bounded *a.s.*, and thus $\mathbb{P}(A) = 1$. Let $\omega \in A$, now we can estimate $\|F(u_k(\omega))\|$ using the $\alpha$-symmetric assumption on the operator.

**Case** $\alpha \in (0, 1)$.

$$\|F(u_k(\omega)) - F(u^*)\| \leq \|u_k(\omega) - v^*\|(K_0 + K_1\|F(v^*)\|^\alpha + K_2\|u_k(\omega) - v^*\|^{\alpha/(1-\alpha)}). \quad (50)$$

Since $\omega \in A$, it follows that for all $k \geq 0$,

$$\|u_k(\omega) - v^*\| \leq C(\omega).$$

Using this fact and equation (50) we obtain that for all $k \geq 0$,

$$\|F(u_k(\omega))\| \leq C(\omega)(K_0 + K_1\|F(v^*)\|^\alpha + K_2 C(\omega)^{\alpha/(1-\alpha)}) + \|F(v^*)\|. \quad (51)$$

Thus, the sequence $\{\|F(u_k(\omega))\|\}$ is upper bounded by some constant $C_1(\omega)$. Where $C_1(\omega) = C(\omega)(K_0 + K_1\|F(v^*)\|^\alpha + K_2 C(\omega)^{\alpha/(1-\alpha)}) + \|F(v^*)\|$.

**Case** $\alpha = 1$.

For $\alpha = 1$ by Proposition 2.2 we have

$$\|F(u_k(\omega)) - F(v^*)\| \leq \|u_k(\omega) - v^*\|(L_0 + L_1\|F(v^*)\|) \exp(L_1\|u_k(\omega) - v^*\|). \quad (52)$$

We can get the following bound for $\|F(u_k(\omega))\|$, for all $k \geq 0$:

$$\begin{aligned}
\|F(u_k(\omega))\| &\leq \|F(u_k(\omega)) - F(v^*)\| + \|F(v^*)\| \\
&\leq \|u_k(\omega) - v^*\|(L_0 + L_1\|F(v^*)\|) \exp(L_1\|u_k(\omega) - v^*\|) + \|F(v^*)\|. \quad (53)
\end{aligned}$$

Since $\omega \in A$, the following bound holds for any $k \geq 0$

$$\|u_k(\omega) - v^*\| \leq C(\omega).$$

Using these facts and equation (53) for all $k \geq 0$:

$$\begin{aligned}
\|F(u_k(\omega))\| &\leq \|u_k(\omega) - v^*\|(L_0 + L_1\|F(v^*)\|) \exp(L_1\|u_k(\omega) - v^*\|) + \|F(v^*)\| \\
&\leq C(\omega)(L_0 + L_1\|F(v^*)\|) \exp(L_1 C(\omega)) + \|F(v^*)\|. \quad (54)
\end{aligned}$$

We showed that for all $k \geq 0$ the norm $\|F(u_k(\omega))\|$ is upper bounded by some constant $\overline{C}_1(\omega)$, where $\overline{C}_1(\omega) = C(\omega)(L_0 + L_1\|F(v^*)\|) \exp(L_1 C(\omega)) + \|F(v^*)\|$. Then, for both cases $\alpha \in (0, 1)$ and $\alpha = 1$ in equations (51), (54) we showed that $\|F(u_k(\omega))\|$ is upper bounded by $\max\{C_1(\omega), \overline{C}_1(\omega)\}$. Using these results and comparison test it follows that: for all $\omega \in A$,

$$\begin{aligned}
\sum_{k=0}^{\infty} \beta_k \min\left\{1, \frac{1}{\|F(u_k(\omega))\| + \sigma}\right\} &\geq \sum_{k=0}^{\infty} \beta_k \min\left\{1, \frac{1}{\max\{C_1(\omega), \overline{C}_1(\omega)\} + \sigma}\right\} \\
&= \min\left\{1, \frac{1}{\max\{C_1(\omega), \overline{C}_1(\omega)\} + \sigma}\right\} \sum_{k=0}^{\infty} \beta_k \\
&= \infty,
\end{aligned} \quad (55)$$

where the last equality holds by the choice of parameters $\beta_k$: $\sum_{i=0}^{\infty} \beta_k = \infty$. Since $\mathbb{P}(A) = 1$, it follows that

$$\mathbb{P}\left(\sum_{k=0}^{\infty} \beta_k \min\left\{1, \frac{1}{\|F(u_k)\| + \sigma}\right\} = \infty\right) = 1.$$

Combining this with (49) we obtain

$$\mathbb{P}\left(\sum_{k=0}^{\infty} \mathbb{E}[\gamma_k \mid \mathcal{F}_{k-1}] = \infty\right) = 1. \quad (56)$$

By Lemma 3.1, we have

$$\sum_{k=0}^{\infty} \mathbb{E}[\gamma_k \mid \mathcal{F}_{k-1}] \operatorname{dist}^p(u_k, U^*) < \infty. \quad (57)$$

Due to $\mathbb{E}[\gamma_k \mid \mathcal{F}_{k-1}] = \infty$ almost surely, it follows that

$$\liminf_{k \to \infty} \text{dist}^p(u_k, U^*) = 0 \qquad a.s. \tag{58}$$

Since $\|u_k - u^*\|$ converges *a.s.* for any given $u^* \in U^*$, the sequence $\{u_k\}$ is bounded *a.s.* and has accumulation points *a.s.* Let $\{k_i\}$ be an index sequence, such that

$$\lim_{i \to \infty} \text{dist}^p(u_{k_i}, U^*) = \liminf_{k \to \infty} \text{dist}^p(u_k, U^*) = 0 \quad a.s. \tag{59}$$

We assume that the sequence $\{u_{k_i}\}$ is convergent with a limit point $\bar{u}$; otherwise, we choose a convergent subsequence. Therefore,

$$\lim_{i \to \infty} \|u_{k_i} - \bar{u}\| = 0 \quad a.s. \tag{60}$$

Then, by (58), $\text{dist}(\bar{u}, U^*) = 0$, thus $\bar{u} \in U^*$ *a.s.* since $U^*$ is closed. Since the sequence $\{\|u_k - u^*\|\}$ converges *a. s.* for all $u^* \in U^*$, by (60) we have

$$\lim_{k \to \infty} \|u_k - \bar{u}\| = 0 \quad a.s. \tag{61}$$

∎

## B.3 PROOF OF LEMMA 3.3

*Proof.* By taking the total expectation in (41) in Lemma 3.1 and using the definition of the stepsize $\gamma_k$, we obtain for any solution $u^* \in U^*$ and all $k \geq 0$,

$$\mathbb{E}[\|u_{k+1} - u^*\|^2] \leq \mathbb{E}[\|u_k - u^*\|^2] - 2\mu\mathbb{E}[\gamma_k \text{dist}^p(u_k, U^*)] + 3\beta_k^2(2\sigma^2 + 1). \tag{62}$$

The equation (62) satisfies the conditions of Lemma A.3 with

$$\bar{v}_k = \mathbb{E}[\|u_k - u^*\|^2], \quad \bar{a}_k = 0, \quad \bar{z}_k = 2\mu\mathbb{E}[\gamma_k \text{dist}^p(u_k, U^*)], \quad \bar{b}_k = 3\beta_k^2(2\sigma^2 + 1). \tag{63}$$

Thus, by Lemma A.3, it follows that the sequence $\{\mathbb{E}[\|u_k - u^*\|^2]\}$ converges to a non-negative scalar for any $u^* \in U^*$. Therefore, the sequence $\{\mathbb{E}[\|u_k - u^*\|^2]\}$ is bounded for all $u^* \in U^*$. Next, using the property of $\alpha$-symmetric operators, we show that $\{\mathbb{E}[\|F(u_k)\|]\}$ is bounded. Let $v^* \in U^*$ be an arbitrary, but fixed solution. Then, by the $\alpha$-symmetric property of $F$, we have that

$$\begin{aligned}
\|F(u_k)\| &\leq \|F(u_k) - F(v^*)\| + \|F(v^*)\| \\
&\leq \|u_k - v^*\|(K_0 + K_1\|F(v^*)\|^\alpha + K_2\|u_k - v^*\|^{\alpha/(1-\alpha)}) + \|F(v^*)\|.
\end{aligned} \tag{64}$$

Taking expectation, we obtain

$$\mathbb{E}[\|F(u_k)\|] \leq (K_0 + K_1\|F(v^*)\|^\alpha)\mathbb{E}[\|u_k - v^*\|] + K_2\mathbb{E}[\|u_k - v^*\|^{1+\alpha/(1-\alpha)}] + \|F(v^*)\|. \tag{65}$$

Notice, that $\mathbb{E}[\|u_k - v^*\|^{1+\alpha/(1-\alpha)}] = \mathbb{E}[(\|u_k - v^*\|^2)^{1/2(1-\alpha)}]$, and for $\alpha \leq 1/2$, the quantity $1/2(1 - \alpha) \leq 1$. Thus, we can apply Jensen inequality for concave function

$$\mathbb{E}[(\|u_k - v^*\|^2)^{1/2(1-\alpha)}] \leq \mathbb{E}[\|u_k - v^*\|^2]^{1/2(1-\alpha)}.$$

Therefore, using these results and Jensen inequality for the first term in equation (65), we obtain

$$\mathbb{E}[\|F(u_k)\|] \leq (K_0 + K_1\|F(v^*)\|^\alpha)\mathbb{E}[\|u_k - v^*\|^2]^{1/2} + K_2\mathbb{E}[\|u_k - v^*\|^2]^{1/2(1-\alpha)} + \|F(v^*)\|. \tag{66}$$

Since $\mathbb{E}[\|u_k - v^*\|^2]$ is bounded, $\mathbb{E}[\|F(u_k)\|]$ is bounded by some constant $C_F > 0$ for all $k \geq 0$. ∎

## B.4 PROOF OF THEOREM 3.4

*Proof.* Letting $y = P_{U^*}(u_k)$ in equation (39) in Lemma 3.1 and using $p$-quasi sharpness we obtain

$$\begin{aligned}
\|u_{k+1} - P_{U^*}(u_k)\|^2 \leq{}& \|u_k - P_{U^*}(u_k)\|^2 - 2\mu\gamma_k\text{dist}^p(u_k, U^*) \\
&+ 2\gamma_k\langle e_k, u_k - P_{U^*}(u_k)\rangle + 3\beta_k^2(\|e_k\|^2 + \|e_k^2\|^2 + 1). \tag{67}
\end{aligned}$$

By the definition of the distance function, we have

$$\text{dist}^2(u_{k+1}, U^*) \leq \|u_{k+1} - P_{U^*}(u_k)\|^2.$$

Thus,

$$\begin{aligned}
\text{dist}^2(u_{k+1}, U^*) \leq \text{dist}^2(u_k, U^*) &- 2\mu\gamma_k\text{dist}^p(u_k, U^*) \\
&+ 2\gamma_k\langle e_k, u_k - P_{U^*}(u_k)\rangle + 3\beta_k^2(\|e_k\|^2 + \|e_k^2\|^2 + 1). \quad (68)
\end{aligned}$$

By Assumption 2.5 and the law of total expectation, and independence of samples $\xi_k$ and $\xi_k^2$, it follows that

$$\begin{aligned}
\mathbb{E}[\gamma_k\langle e_k, u_k - P_{U^*}(u_k)\rangle] &= \mathbb{E}[\mathbb{E}[\gamma_k\langle e_k, u_k - P_{U^*}(u_k)\rangle \mid \mathcal{F}_{k-1}]] \\
&= \mathbb{E}[\mathbb{E}[\gamma_k \mid \mathcal{F}_{k-1}]\langle\mathbb{E}[e_k \mid \mathcal{F}_{k-1}], u_k - P_{U^*}(u_k)\rangle] \\
&= 0. \quad (69)
\end{aligned}$$

Also, we have $\mathbb{E}[\mathbb{E}[\|e_k^1\| \mid \mathcal{F}_{k-1}] \leq \sigma^2$ and $\mathbb{E}[\mathbb{E}[\|e_k^2\| \mid \mathcal{F}_{k-1}] \leq \sigma^2$. Thus, by taking the total expectation in (68), we obtain

$$\mathbb{E}[\text{dist}^2(u_{k+1}, U^*)] \leq \mathbb{E}[\text{dist}^2(u_k, U^*)] - 2\mu\mathbb{E}[\gamma_k\text{dist}^p(u_k, U^*)] + 3\beta_k^2(2\sigma^2 + 1). \quad (70)$$

We aim to upper bound $2\mu\mathbb{E}[\gamma_k\text{dist}^p(u_k, U^*)]$. To do so consider an event $A_k$, defined as follows:

$$A_k = \{\|F(u_k)\| + \|e_k\| \leq 2(\mathbb{E}[\|F(u_k)\|] + \mathbb{E}[\|e_k\|])\}.$$

Then, by the law of total expectation, we obtain

$$\mathbb{E}[\gamma_k\text{dist}^p(u_k, U^*)] = \mathbb{E}[\gamma_k\text{dist}^p(u_k, U^*)|A_k]\mathbb{P}(A_k) + \mathbb{E}[\gamma_k\text{dist}^p(u_k, U^*)|\overline{A}_k]\mathbb{P}(\overline{A}_k), \quad (71)$$

where $\overline{A}$ denotes the complement of an event $A$. We want to provide a lower bound on $\mathbb{P}(A_k)$. To do so, we upperbound $\mathbb{P}(\overline{A}_k)$ using Markov's inequality, as follows:

$$\begin{aligned}
\mathbb{P}(\overline{A}_k) &= \mathbb{P}\left(\{\|F(u_k)\| + \|e_k\| > 2(\mathbb{E}[\|F(u_k)\|] + \mathbb{E}[\|e_k\|])\}\right) \\
&\leq \frac{\mathbb{E}[\|F(u_k)\|] + \mathbb{E}[\|e_k\|]}{2(\mathbb{E}[\|F(u_k)\|] + \mathbb{E}[\|e_k\|])} \\
&= \frac{1}{2}. \quad (72)
\end{aligned}$$

Thus,

$$\begin{aligned}
\mathbb{E}[\gamma_k\text{dist}^p(u_k, U^*)] &= \mathbb{E}[\gamma_k\text{dist}^p(u_k, U^*)|A_k](1 - \mathbb{P}(\overline{A}_k)) + \mathbb{E}[\gamma_k\text{dist}^p(u_k, U^*)|\overline{A}_k]\mathbb{P}(\overline{A}_k) \\
&\geq \frac{1}{2}\mathbb{E}[\gamma_k\text{dist}^p(u_k, U^*)|A_k] + \mathbb{E}[\gamma_k\text{dist}^p(u_k, U^*)|\overline{A}_k]\mathbb{P}(\overline{A}_k) \\
&\geq \frac{1}{2}\mathbb{E}[\gamma_k\text{dist}^p(u_k, U^*)|A_k]. \quad (73)
\end{aligned}$$

By the definition of the event $A_k$, we have

$$\begin{aligned}
\mathbb{E}[\gamma_k\text{dist}^p(u_k, U^*)|A_k] &= \beta_k\mathbb{E}\left[\min\left\{1, \frac{1}{\|\Phi(u_k, \xi_k)\|}\right\}\text{dist}^p(u_k, U^*)|A_k\right] \\
&\geq \beta_k\mathbb{E}\left[\min\left\{1, \frac{1}{\|F(u_k)\| + \|e_k\|}\right\}\text{dist}^p(u_k, U^*)|A_k\right] \\
&\geq \beta_k\min\left\{1, \frac{1}{2(\mathbb{E}[\|F(u_k)\|] + \mathbb{E}[\|e_k\|])}\right\}\mathbb{E}[\text{dist}^p(u_k, U^*)|A_k]. \quad (74)
\end{aligned}$$

By Lemma 3.3, $\mathbb{E}[\|F(u_k)\|] \leq C_F$ for all $k \geq 0$, and by Assumption 2.5 and Jensen inequality, we have $\mathbb{E}[\|e_k\|] \leq \mathbb{E}[\|e_k\|^2]^{1/2} \leq \sigma$. Thus, it follows that

$$\mathbb{E}[\gamma_k\text{dist}^p(u_k, U^*)] \geq \frac{1}{2}\beta_k\min\left\{1, \frac{1}{2(C_F + \sigma)}\right\}\mathbb{E}[\text{dist}^p(u_k, U^*)]. \quad (75)$$

Combining equations (70) and (75), and using $a = \mu\min\left\{1, \frac{1}{2(C_F+\sigma)}\right\}$, we obtain

$$\mathbb{E}[\text{dist}^2(u_{k+1}, U^*)] \leq \mathbb{E}[\text{dist}^2(u_k, U^*)] - a\beta_k\mathbb{E}[\text{dist}^p(u_k, U^*)] + 3\beta_k^2(2\sigma^2 + 1). \quad (76)$$

Now let $D_k = \mathbb{E}[\text{dist}^2(u_k, U^*)]$, and consider the following two cases:

**Case $p = 2$.** When $p = 2$, equation (76) satisfies the assumptions of Lemma A.6 with

$$r_k = D_k, \quad \alpha_k = \beta_k, \quad s_k = 0, \quad d = a, \quad c = 3(2\sigma^2 + 1). \tag{77}$$

Then, by Lemma A.6, we get the following convergence rate for all $k \geq 1$,

$$D_{k+1} \leq \frac{8D_0}{k^2} + \frac{6(2\sigma^2 + 1)}{a^2 k}. \tag{78}$$

**Case $p > 2$.** When $p \geq 2$, By applying telescoping sum to inequality (76) and rearranging the terms NEW
we obtain

$$\mathbb{E}[a \sum_{t=0}^{k} \beta_k \text{dist}^p(u_k, U^*)] \leq D_0 - D_{k+1} + 3(2\sigma^2 + 1) \sum_{t=0}^{k} \beta_k^2. \tag{79}$$

Since $p \geq 2$, the function $\text{dist}^p(\cdot, U^*)$ is convex, thus by defining $\bar{u}_k = (\sum_{t=0}^{k} \beta_k)^{-1} \sum_{t=0}^{k} \beta_k u_t$
and applying Jensen inequality be obtain

$$(\sum_{t=0}^{k} \beta_k)\mathbb{E}[\text{dist}^p(\bar{u}_k, U^*)] \leq \mathbb{E}[\sum_{t=0}^{k} \beta_k \text{dist}^p(u_k, U^*)].$$

Since $p \geq 2$, by applying Jensen inequality one more time, we obtain

$$(\bar{D}_k)^{p/2} = \left(\mathbb{E}[\text{dist}^2(\bar{u}_k, U^*)]\right)^{p/2} \leq \mathbb{E}\left[\left(\text{dist}^2(\bar{u}_k, U^*)\right)^{p/2}\right] = \mathbb{E}[\text{dist}^p(\bar{u}_k, U^*)].$$

Applying these estimates, we get

$$(\bar{D}_k)^{p/2} \sum_{t=0}^{k} \beta_t \leq \sum_{t=0}^{k} \beta_t D_t^{p/2} \leq \frac{1}{a}\left(D_0 - D_{k+1} + 3(2\sigma^2 + 1)\sum_{t=0}^{k}\beta_t^2\right). \tag{80}$$

Since $\beta_k = \frac{b}{(k+1)^q}$, with $b > 0, 1 > q > 1/2$, then $\{\beta_k\}$ satisfies the conditions of Lemma 3.3. Also,
by Lemma A.7 the following inequalities hold: for all $k \geq 1$,

$$\sum_{t=0}^{k} \beta_t \geq \frac{b}{1-q}((k+1)^{1-q} - 2^{1-q}), \qquad \sum_{t=0}^{k} \beta_t^2 \leq \frac{b^2}{2q-1}. \tag{81}$$

Combining equations (80) and (81), and omitting $D_{k+1}$, we obtain

$$(\bar{D}_k)^{p/2} \leq \frac{(1-q)\left(D_0 + 3b^2(2\sigma^2 + 1)/(2q-1)\right)}{ab\left((k+1)^{1-q} - 2^{1-q}\right)}. \tag{82}$$

Raising both sides of the preceding inequality in power $2/p$, we obtain

$$\bar{D}_k \leq \frac{(1-q)^{2/p}\left(D_0 + 3b^2(2\sigma^2 + 1)/(2q-1)\right)^{2/p}}{(ab)^{2/p}\left((k+1)^{1-q} - 2^{1-q}\right)^{2/p}}. \tag{83}$$

∎

## C  KORPELEVICH METHOD ANALYSIS

**Lemma C.1.** *Let $U$ be a closed convex set. Then, for the iterate sequences $\{u_k\}$ and $\{h_k\}$ generated by the stochastic Korpelevich method (8) and $y \in U$ and $k \geq 0$,*

$$\|h_{k+1} - y\|^2 \leq \|h_k - y\|^2 - \|h_k - u_k\|^2 - 2\gamma_k\langle F(u_k), u_k - y\rangle - 2\gamma_k\langle e_k^2, u_k - y\rangle$$
$$+ 3\gamma_k^2\|F(h_k) - F(u_k)\|^2 + 3\gamma_k^2(\|e_k^2\|^2 + \|e_k^1\|^2),$$

*where $e_k^1 = \Phi(h_k, \xi_k^1) - F(h_k)$, $e_k^2 = \Phi(u_k, \xi_k^2) - F(u_k)$ for all $k \geq 0$.*

*Proof.* Let $k \geq 0$ be arbitrary but fixed. By the definition of $h_{k+1}$ in (8), we have $\|h_{k+1} - y\| = \|P_U(h_k - \gamma_k \Phi(u_k, \xi_k^2)) - y\|$ for any $y \in U$. Using the projection inequality, we obtain for any $y \in U$,

$$\|h_{k+1} - y\|^2 \leq \|h_k - \gamma_k \Phi(u_k, \xi_k^2) - y\|^2 - \|h_{k+1} - h_k + \gamma_k \Phi(u_k, \xi_k^2)\|^2$$
$$\leq \|h_k - y\|^2 - \|h_{k+1} - h_k\|^2 + 2\gamma_k \langle \Phi(u_k, \xi_k^2), y - h_{k+1} \rangle. \tag{84}$$

Next, we consider the term $\|h_{k+1} - h_k\|^2$, where we add and subtract $u_k$, thus

$$\|h_{k+1} - h_k\|^2 = \|h_{k+1} - u_k\|^2 + \|h_k - u_k\|^2 - 2\langle h_{k+1} - u_k, h_k - u_k \rangle. \tag{85}$$

Adding and subtracting $2\gamma_k \langle \Phi(h_k, \xi_k^1), u_k - h_{k+1} \rangle$, and combining (84) and (85) we obtain

$$\|h_{k+1} - y\|^2 \leq \|h_k - y\|^2 - \|h_{k+1} - u_k\|^2 - \|h_k - u_k\|^2 + 2\langle h_{k+1} - u_k, h_k - u_k \rangle$$
$$+ 2\gamma_k \langle \Phi(u_k, \xi_k^2), y - u_k + u_k - h_{k+1} \rangle + 2\gamma_k \langle \Phi(h_k, \xi_k^1) - \Phi(h_k, \xi_k^1), u_k - h_{k+1} \rangle$$
$$\leq \|h_k - y\|^2 - \|h_{k+1} - u_k\|^2 - \|h_k - u_k\|^2 + 2\langle h_{k+1} - u_k, h_k - \gamma_k \Phi(h_k, \xi_k^1) - u_k \rangle$$
$$+ 2\gamma_k \langle \Phi(u_k, \xi_k^2), y - u_k \rangle + 2\gamma_k \langle \Phi(h_k, \xi_k^1) - \Phi(u_k, \xi_k^2), h_{k+1} - u_k \rangle. \tag{86}$$

Since $u_k = P_U(h_k - \gamma_k \Phi(h_k, \xi_k^1))$ and $h_{k+1} \in U$, by the projection inequality in (25), it follows that

$$2\langle h_{k+1} - u_k, h_k - \gamma_k \Phi(h_k, \xi_k^1) - u_k \rangle \leq 0.$$

Using Cauchy-Schwarz inequality and relation $2ab \leq a^2 + b^2$ for $a, b \in \mathbb{R}$, we obtain

$$2\gamma_k \langle \Phi(h_k, \xi_k^1) - \Phi(u_k, \xi_k^2), h_{k+1} - u_k \rangle \leq 2\gamma_k \|\Phi(h_k, \xi_k^1) - \Phi(u_k, \xi_k^2)\| \|h_{k+1} - u_k\|$$
$$\leq \gamma_k^2 \|\Phi(h_k, \xi_k^1) - \Phi(u_k, \xi_k^2)\|^2 + \|h_{k+1} - u_k\|^2.$$

Using triangle inequality and relation $(\sum_{i=1}^m a_i)^2 \leq m \sum_{i=1}^m a_i^2$ we get

$$\|\Phi(h_k, \xi_k^1) - \Phi(u_k, \xi_k^2)\|^2 = \|\Phi(h_k, \xi_k^1) - F(h_k) + F(h_k) - F(u_k) + F(u_k) - \Phi(u_k, \xi_k^2)\|^2$$
$$\leq 3(\|e_k^1\|^2 + \|F(h_k) - F(u_k)\|^2 + \|e_k^2\|^2).$$

Combining the preceding three estimates with (86), we get the stated relation

$$\|h_{k+1} - y\|^2 \leq \|h_k - y\|^2 - \|h_k - u_k\|^2 - 2\gamma_k \langle F(u_k), u_k - y \rangle - 2\gamma_k \langle e_k^2, u_k - y \rangle$$
$$+ 3\gamma_k^2 \|F(h_k) - F(u_k)\|^2 + 3\gamma_k^2(\|e_k^2\|^2 + \|e_k^1\|^2).$$

∎

## C.1 Proof of Lemma 4.1

*Proof.* By Lemma C.1 we have for all $k \geq 0$ and for all $y \in U$,

$$\|h_{k+1} - y\|^2 \leq \|h_k - y\|^2 - \|h_k - u_k\|^2 - 2\gamma_k \langle F(u_k), u_k - y \rangle - 2\gamma_k \langle e_k^2, u_k - y \rangle$$
$$+ 3\gamma_k^2 \|F(h_k) - F(u_k)\|^2 + 3\gamma_k^2(\|e_k^2\|^2 + \|e_k^1\|^2), \tag{87}$$

with $e_k^1 = \Phi(h_k, \xi_k^1) - F(h_k)$ and $e_k^2 = \Phi(u_k, \xi_k^2) - F(u_k)$ for all $k \geq 0$. We want to estimate the term $\|F(h_k) - F(h_{k-1})\|^2$ on the LHS of the inequality using the fact that $F(\cdot)$ is an $\alpha$-symmetric operator for two cases **(a)** $\alpha \in (0, 1)$ and **(b)** $\alpha = 1$.

**Case** $\alpha \in (0, 1)$. Using the alternative characterization of $\alpha$-symmetric operators from Proposition 2.2(a) (as given in (4)), when $\alpha \in (0, 1)$, the next inequality holds for any $k \geq 0$,

$$\|F(h_k) - F(u_k)\| \leq \|h_k - u_k\|(K_0 + K_1\|F(h_k)\|^\alpha + K_2\|h_k - u_k\|^{\alpha/(1-\alpha)}). \tag{88}$$

We want to separate $\|F(h_k)\|$ into two parts: stochastic approximation of operator $\Phi(h_k, \xi_k^1)$ and error $e_k^1$. Recall that $e_k^1 = F(h_k) - \Phi(h_k, \xi_k^1)$, then based on triangle inequality $\|F(h_k)\| \leq \|\Phi(h_k, \xi_k^1)\| + \|e_k^1\|$, and since $\alpha \leq 1$ we obtain

$$\|F(h_k)\|^\alpha \leq \|\Phi(h_k, \xi_k^1)\|^\alpha + \|e_k^1\|^\alpha. \tag{89}$$

Thus, combining this fact with (88) we get the following estimation

$$\|F(h_k) - F(u_k)\| \leq \|h_k - u_k\|(K_0 + K_1\|\Phi(h_k, \xi_k^1)\|^\alpha + K_1\|e_k^1\|^\alpha + K_2\|h_k - u_k\|^{\alpha/(1-\alpha)}). \tag{90}$$

By the projection property (26) and the stepsize choice (16), we have

$$\|h_k - u_k\| \leq \gamma_k\|\Phi(h_k, \xi_k^1)\| = \beta_k \min\{1, \frac{1}{\|\Phi(h_k, \xi_k^1)\|}\}\|\Phi(h_k, \xi_k^1)\| \leq \beta_k \leq 1. \tag{91}$$

Then, $K_2\|h_k - u_k\|^{\alpha/(1-\alpha)} \leq K_2$, and

$$\gamma_k\|F(h_k) - F(u_k)\| \leq \gamma_k(K_0 + K_1\|\Phi(h_k, \xi_k^1)\|^\alpha + K_1\|e_k^1\|^\alpha + K_2)\|h_k - u_k\|$$

$$\leq \beta_k(K_0 \min\{1, \frac{1}{\|\Phi(h_k, \xi_k^1)\|}\} + K_1 \min\{1, \frac{1}{\|\Phi(h_k, \xi_k^1)\|}\}\|\Phi(h_k, \xi_k^1)\|^\alpha)\|h_k - u_k\|$$

$$+ \beta_k(K_1\|e_k^1\|^\alpha + K_2 \min\{1, \frac{1}{\|\Phi(h_k, \xi_k^1)\|}\})\|h_k - u_k\|$$

$$\leq \beta_k(K_0 + K_1 + K_2)\|h_k - u_k\| + \beta_k K_1\|e_k^1\|^\alpha\|h_k - u_k\|. \tag{92}$$

By inequality (91), we have $\|h_k - u_k\| \leq 1$, and using this estimate in equation (92) we obtain

$$\gamma_k\|F(h_k) - F(u_k)\| \leq \beta_k(K_0 + K_1 + K_2)\|h_k - u_k\| + \beta_k K_1\|e_k^1\|^\alpha. \tag{93}$$

**Case** $\alpha = 1$. Based on the alternative characterization of $\alpha$-symmetric operators from Proposition 2.2(b) (as given in (16)), when $\alpha = 1$, the following inequality holds for any $k \geq 0$,

$$\|F(h_k) - F(u_k)\| \leq \|h_k - u_k\|(L_0 + L_1\|\Phi(h_k, \xi_k^1)\|)\exp(L_1\|h_k - u_k\|). \tag{94}$$

We upperbound $\|F(h_k)\|$ using equation (89) and get

$$\|F(h_k) - F(u_k)\| \leq \|h_k - u_k\|(L_0 + L_1\|F(h_k)\| + L_1\|e_k^1\|)\exp(L_1\|h_k - u_k\|). \tag{95}$$

Note that relation in (91) holds irrespective of the value of $\alpha$. Thus, since $\|h_k - u_k\| \leq 1$, we have $\exp(L_1\|h_k - u_k\|) \leq \exp(L_1\beta_k)$, and we obtain

$$\gamma_k\|F(h_k) - F(u_k)\| \leq \gamma_k(L_0 + L_1\|\Phi(h_k, \xi_k^1)\| + L_1\|e_k^1\|)\exp(L_1\beta_k)\|h_k - u_k\|$$

$$= \exp(L_1\beta_k)L_0\beta_k \min\{1, \frac{1}{\|\Phi(h_k, \xi_k^1)\|}\}\|h_k - u_k\|$$

$$+ \exp(L_1\beta_k)L_1\beta_k \min\{1, \frac{1}{\|\Phi(h_k, \xi_k^1)\|}\}\|\Phi(h_k, \xi_k^1)\|\|h_k - u_k\|$$

$$+ \exp(L_1\beta_k)L_1\beta_k \min\{1, \frac{1}{\|\Phi(h_k, \xi_k^1)\|}\}\|e_k^1\|\|h_k - u_k\|$$

$$\leq \exp(L_1\beta_k)\beta_k(L_0 + L_1 + L_1\|e_k^1\|)\|h_k - u_k\|. \tag{96}$$

By inequality (91), we have $\|h_k - u_k\| \leq 1$. Using this estimate in (96), we furher obtain

$$\gamma_k\|F(h_k) - F(u_k)\| \leq \exp(L_1\beta_k)\beta_k(L_0 + L_1)\|h_k - u_k\| + \exp(L_1\beta_k)\beta_k\|e_k^1\|. \tag{97}$$

Now, we are done with the cases of $\alpha$ values. Let

$$C_a(\beta_k, \alpha) = \begin{cases} (K_0 + K_1 + K_2), & \text{when } \alpha \in (0, 1), \\ \exp(L_1\beta_k)(L_0 + L_1), & \text{when } \alpha = 1. \end{cases} \tag{98}$$

Also, define

$$C_e(\beta_k, \alpha) = \begin{cases} K_1, & \text{when } \alpha \in (0, 1), \\ \exp(L_1\beta_k), & \text{when } \alpha = 1. \end{cases} \tag{99}$$

Then, by inequality $(\sum_{i=1}^m a_i)^2 \leq m\sum_{i=1}^m a_i^2$, for both cases we have

$$\gamma_k^2\|F(h_k) - F(u_k)\|^2 \leq 2\beta_k^2 C_a(\beta_k, \alpha)^2\|h_k - u_k\|^2 + 2\beta_k^2 C_e(\beta_k, \alpha)^2\|e_k^1\|^{2\alpha}. \tag{100}$$

Combining preceding inequality with (87) we obtain that for any $k \geq 0$,

$$\|h_{k+1} - y\|^2 \leq \|h_k - y\|^2 - (1 - 6\beta_k^2 C_a(\beta_k, \alpha)^2)\|h_k - u_k\|^2 - 2\gamma_k \langle F(u_k), u_k - y \rangle$$
$$- 2\gamma_k \langle e_k^2, u_k - y \rangle + 6\beta_k^2 C_e(\beta_k, \alpha)^2 \|e_k^1\|^{2\alpha} + 3\gamma_k^2 (\|e_k^2\|^2 + \|e_k^1\|^2). \tag{101}$$

Next, we plug $y = u^*$, where $u^* \in U^*$ is an arbitrary solution and use $p$-quasi sharpness of the operator $F$ to obtain

$$\|h_{k+1} - u^*\|^2 \leq \|h_k - u^*\|^2 - (1 - 6\beta_k^2 C_a(\beta_k, \alpha)^2)\|h_k - u_k\|^2 - 2\gamma_k \mu \mathrm{dist}^p(u_k, U^*)$$
$$- 2\gamma_k \langle e_k^2, u_k - u^* \rangle + 6\beta_k^2 C_e(\beta_k, \alpha)^2 \|e_k^1\|^{2\alpha} + 3\gamma_k^2 (\|e_k^2\|^2 + \|e_k^1\|^2). \tag{102}$$

By the stepsize choice $\gamma_k \leq \beta_k$, thus

$$\|h_{k+1} - u^*\|^2 \leq \|h_k - u^*\|^2 - (1 - 6\beta_k^2 C_a(\beta_k, \alpha)^2)\|h_k - u_k\|^2 - 2\gamma_k \mu \mathrm{dist}^p(u_k, U^*)$$
$$- 2\gamma_k \langle e_k^2, u_k - u^* \rangle + 6\beta_k^2 C_e(\beta_k, \alpha)^2 \|e_k^1\|^{2\alpha} + 3\beta_k^2 (\|e_k^2\|^2 + \|e_k^1\|^2). \tag{103}$$

Since $\sum_{k=0}^{\infty} \beta_k^2 < \infty$, it follows that $\beta_k \to 0$. By definitions of $C_a(\beta_k, \alpha)$ and $C_e(\beta_k, \alpha)$ in (98) and (99), respectively, there exists $N \geq 0$ such that the stepsizes satisfy $1 - 6\beta_k^2 C_a(\beta_k, \alpha)^2 \geq \frac{1}{2}$ and $C_e(\beta_k, \alpha)^2 \leq \max\{K_1, \exp(L_1, \beta_k)\} \leq \max\{K_1, 2\}$. Thus, the following inequality holds for any $k \geq N$,

$$\|h_{k+1} - u^*\|^2 \leq \|h_k - u^*\|^2 - \frac{1}{2}\|h_k - u_k\|^2 - 2\gamma_k \mu \mathrm{dist}^p(u_k, U^*)$$
$$- 2\gamma_k \langle e_k^2, u_k - u^* \rangle + 3\beta_k^2 (\|e_k^2\|^2 + \|e_k^1\|^2 + 2\max\{K_1, 2\}\|e_k^1\|^{2\alpha}). \tag{104}$$

Recalling that $e_k^1 = \Phi(h_k, \xi_k^1) - F(h_k)$, $e_k^2 = \Phi(u_k, \xi_k^2) - F(u_k)$ and using the stochastic properties of $\xi_k^1, \xi_k^2$ imposed by Assumption 2.5 and method's updates, we have

$$\mathbb{E}[\gamma_k \langle e_k^2, u_k - u^* \rangle \mid \mathcal{F}_{k-1}] = \mathbb{E}[\gamma_k \mathbb{E}[\langle e_k^2, u_k - u^* \rangle \mid \mathcal{F}_{k-1} \cup \{\xi_k^1\}] \mid \mathcal{F}_{k-1}] = 0,$$

since stepsize $\gamma_k$ is measurable in $\mathcal{F}_{k-1} \cup \{\xi_k^1\}$. Also, it holds that for all $k \geq 0$,

$$\mathbb{E}[\mathbb{E}[\|e_k^2\|^2 \mid \mathcal{F}_{k-1} \cup \{\xi_k^1\}]|\mathcal{F}_{k-1}] \leq \sigma^2, \quad \text{and} \quad \mathbb{E}[\|e_k^1\|^2 \mid \mathcal{F}_{k-1}] \leq \sigma^2.$$

Moreover, since $\alpha \leq 1$, the conditional expectation $\mathbb{E}[\|e_k^1\|^{2\alpha}|\mathcal{F}_{k-1}]$ is finite, and by Jensen inequality, it follows that for all $k \geq 0$,

$$\mathbb{E}[\|e_k^1\|^{2\alpha}|\mathcal{F}_{k-1}] \leq \sigma^{2\alpha}.$$

Therefore, by taking the conditional expectation on $\mathcal{F}_{k-1}$ in relation (104), we obtain for all $u^* \in U^*$ and for all $k \geq N$,

$$\mathbb{E}[\|h_{k+1} - u^*\|^2|\mathcal{F}_{k-1}] \leq \|h_k - u^*\|^2 - \frac{1}{2}\|h_k - u_k\|^2 - 2\mathbb{E}[\gamma_k \mid \mathcal{F}_{k-1}]\mu\mathrm{dist}^p(u_k, U^*)$$
$$+ 6\beta_k^2(\sigma^2 + \max\{K_1, 2\}\sigma^{2\alpha}). \tag{105}$$

By Lemma A.2, it follows that the sequence $\{\|h_k - u^*\|^2\}$ converges *a.s.* to a non-negative scalar for any $u^* \in U^*$, and almost surely we have

$$\sum_{k=0}^{\infty} \mathbb{E}[\gamma_k \mid \mathcal{F}_{k-1}] \mathrm{dist}^p(u_k, U^*) < \infty, \quad \sum_{k=0}^{\infty} \|h_k - u_k\|^2 < \infty. \tag{106}$$

Since the sequence $\{\|h_k - u^*\|^2\}$ converges *a.s.* for all $u^* \in U^*$, it follows that the sequence $\{\|h_k - u^*\|\}$ is bounded *a.s.* for all $u^* \in U^*$. ∎

### C.2 PROOF OF THEOREOM 4.2

*Proof.* By Lemma 4.1, we almost surely have

$$\sum_{k=0}^{\infty} \mathbb{E}[\gamma_k \mid \mathcal{F}_{k-1}] \mathrm{dist}^p(u_k, U^*) < \infty. \tag{107}$$

The structure of the stepsizes $\gamma_k$ (16) for the Korpelevich method is similar to the stepsizes sequence for the projection method. Following the proof of Theorem 3.2 we obtain

$$\sum_{k=0}^{\infty} \mathbb{E}[\gamma_k \mid \mathbb{F}_{k-1}] = \infty \quad a.s. \tag{108}$$

We provide the full proof of this result below, in spite of the fact it is equivalent to the proof of the same result in Theorem 3.2.

Firstly, we provide a sequence of lower bounds on $\mathbb{E}[\gamma_k \mid \mathcal{F}_{k-1}]$. First, consider the following event:

$$A_k = \{\|e_k^1\| \leq 2\mathbb{E}[\|e_k^1\| \mid \mathcal{F}_{k-1}]\},$$

where $e_k^1 = F(h_k) - \Phi(h_k, \xi_k^1)$ is a stochastic error from the sample for the clipping stepsize $\gamma_k$. Then, by the law of total expectation

$$\mathbb{E}[\gamma_k \mid \mathcal{F}_{k-1}] = \mathbb{E}[\gamma_k \mid \mathcal{F}_{k-1} \cup A_k]\mathbb{P}(A_k \mid \mathcal{F}_{k-1}) + \mathbb{E}[\gamma_k \mid \mathcal{F}_{k-1} \cup \overline{A}_k]\mathbb{P}(\overline{A}_k \mid \mathcal{F}_{k-1}). \tag{109}$$

We want to provide a lower bound on $\mathbb{P}(A_k \mid \mathcal{F}_{k-1})$, to do so, we upperbound $\mathbb{P}(\overline{A}_k \mid \mathcal{F}_{k-1})$ using Markov's inequality and Assumption 2.5:

$$\mathbb{P}(\overline{A}_k \mid \mathcal{F}_{k-1}) = \mathbb{P}(\|e_k^1\| > 2\mathbb{E}[\|e_k^1\| \mid \mathcal{F}_{k-1}]\}) \leq \frac{\mathbb{E}[\|e_k^1\| \mid \mathcal{F}_{k-1}]}{2\mathbb{E}[\|e_k^1\| \mid \mathcal{F}_{k-1}])} = \frac{1}{2}. \tag{110}$$

Thus,

$$\mathbb{E}[\gamma_k \mid \mathcal{F}_{k-1}] = \mathbb{E}[\gamma_k \mid \mathcal{F}_{k-1} \cup A_k](1 - \mathbb{P}(\overline{A}_k \mid \mathcal{F}_{k-1})) + \mathbb{E}[\gamma_k \mid \mathcal{F}_{k-1} \cup \overline{A}_k]\mathbb{P}(\overline{A}_k \mid \mathcal{F}_{k-1})$$

$$\geq \frac{1}{2}\mathbb{E}[\gamma_k \mid \mathcal{F}_{k-1} \cup A_k] + \mathbb{E}[\gamma_k \mid \mathcal{F}_{k-1} \cup \overline{A}_k]\mathbb{P}(\overline{A}_k \mid \mathcal{F}_{k-1})$$

$$\geq \frac{1}{2}\mathbb{E}[\gamma_k \mid \mathcal{F}_{k-1} \cup A_k]. \tag{111}$$

By the definition of $\gamma_k$ and the triangle inequality, we have

$$\mathbb{E}[\gamma_k \mid \mathcal{F}_{k-1} \cup A_k] = \beta_k \mathbb{E}[\min\left\{1, \frac{1}{\|\Phi(h_k, \xi_k^2)\|}\right\} \mid \mathcal{F}_{k-1} \cup A_k]$$

$$\geq \beta_k \mathbb{E}\left[\min\left\{1, \frac{1}{\|F(h_k)\| + \|e_k^2\|}\right\} \mid \mathcal{F}_{k-1} \cup A_k\right]. \tag{112}$$

Next, we use the definition of the event $A_k$ and obtain the following lower bound:

$$\mathbb{E}[\gamma_k \mid \mathcal{F}_{k-1} \cup A_k] \geq \beta_k \mathbb{E}[\min\left\{1, \frac{1}{\|F(h_k)\| + 2\mathbb{E}[\|e_k^2\| \mid \mathcal{F}_{k-1}]}\right\} \mid \mathcal{F}_{k-1} \cup A_k]$$

$$\geq \beta_k \mathbb{E}[\min\left\{1, \frac{1}{\|F(h_k)\| + 2\sigma}\right\} \mid \mathcal{F}_{k-1} \cup A_k]$$

$$= \beta_k \min\left\{1, \frac{1}{\|F(h_k)\| + 2\sigma}\right\}. \tag{113}$$

The first inequality in the preceding equation holds by the definition of the event $A_k = \{\|e_k^2\| \leq 2\mathbb{E}[\|e_k^2\| \mid \mathcal{F}_{k-1}]\}$, while the second inequality holds due to Assumption 2.5 on the $\xi$-samples and Jensen inequality, $\mathbb{E}[\|e_k^2\| \mid \mathcal{F}_{k-1}] \leq \sqrt{\mathbb{E}[\|e_k^2\|^2 \mid \mathcal{F}_{k-1}]} \leq \sigma$. Hence, we have that

$$\sum_{k=0}^{\infty} \frac{1}{2}\beta_k \min\left\{1, \frac{1}{\|F(h_k)\| + 2\sigma}\right\} \leq \sum_{k=0}^{\infty} \mathbb{E}[\gamma_k \mid \mathcal{F}_{k-1}]. \tag{114}$$

Now, we want to show that $\sum_{k=0}^{\infty} \beta_k \min\left\{1, \frac{1}{\|F(h_k)\|+2\sigma}\right\}$ is not summable almost surely, i.e. $\mathbb{P}(\sum_{k=0}^{\infty} \beta_k \min\left\{1, \frac{1}{\|F(h_k)\|+2\sigma}\right\} = \infty) = 1$. We will do so by showing *a.s.* boundedness of

$\|F(h_k)\|$ for all $k \geq 0$, using property of $\alpha$-symmetric operators. To estimate $\|F(h_k)\|$, we add and subtract $F(v^*)$, where $v^* \in U^*$ is an arbitrary but fixed solution, and get

$$\|F(h_k)\| = \|F(h_k) - F(v^*) + F(v^*)\| \leq \|F(h_k) - F(v^*)\| + \|F(v^*)\|.$$

Define the following event:

$$A = \{\omega \in \Omega : \exists\, C(\omega) \in \mathbb{R} \text{ s.t.} \|h_k(\omega) - v^*\| < C(\omega)\ \forall\, k \geq 0\}.$$

Based on Lemma 4.1, the sequence $\{\|h_k - v^*\|\}$ is bounded *a.s.*, and thus $\mathbb{P}(A) = 1$. Let $\omega \in A$, now we can estimate $\|F(h_k(\omega))\|$ using the $\alpha$-symmetric assumption on the operator.

**Case $\alpha \in (0, 1)$.**

$$\|F(h_k(\omega)) - F(u^*)\| \leq \|h_k(\omega) - v^*\|(K_0 + K_1\|F(v^*)\|^\alpha + K_2\|h_k(\omega) - v^*\|^{\alpha/(1-\alpha)}). \quad (115)$$

Since $\omega \in A$, it follows that $\|h_k(\omega) - v^*\| \leq C(\omega)$ for all $k \geq 0$. Using this fact and (115) we obtain that for all $k \geq 0$,

$$\|F(h_k(\omega))\| \leq C(\omega)(K_0 + K_1\|F(v^*)\|^\alpha + K_2 C(\omega)^{\alpha/(1-\alpha)}) + \|F(v^*)\|. \quad (116)$$

Therefore, the sequence $\{\|F(h_k(\omega))\|\}$ is upper bounded by $C_1(\omega) = C(\omega)(K_0 + K_1\|F(v^*)\|^\alpha + K_2 C(\omega)^{\alpha/(1-\alpha)}) + \|F(v^*)\|$.

**Case $\alpha = 1$.**

For $\alpha = 1$ by Proposition 2.2 we have

$$\|F(h_k(\omega)) - F(v^*)\| \leq \|h_k(\omega) - v^*\|(L_0 + L_1\|F(v^*)\|)\exp(L_1\|h_k(\omega) - v^*\|). \quad (117)$$

Therefore, for all $k \geq 0$,

$$\begin{aligned}\|F(h_k(\omega))\| &\leq \|F(h_k(\omega)) - F(v^*)\| + \|F(v^*)\| \\ &\leq \|h_k(\omega) - v^*\|(L_0 + L_1\|F(v^*)\|)\exp(L_1\|h_k(\omega) - v^*\|) + \|F(v^*)\|. \quad (118)\end{aligned}$$

Since $\omega \in A$, we have $\|h_k(\omega) - v^*\| \leq C(\omega)$ for all $k \geq 0$, which when used in (118), implies that for all $k \geq 0$,

$$\begin{aligned}\|F(h_k(\omega))\| &\leq \|h_k(\omega) - v^*\|(L_0 + L_1\|F(v^*)\|)\exp(L_1\|h_k(\omega) - v^*\|) + \|F(v^*)\| \\ &\leq C(\omega)(L_0 + L_1\|F(v^*)\|)\exp(L_1 C(\omega)) + \|F(v^*)\|. \quad (119)\end{aligned}$$

Hence, the sequence $\{\|F(h_k(\omega))\|\}$ is upper bounded by $\overline{C}_1(\omega)$, where $\overline{C}_1(\omega) = C(\omega)(L_0 + L_1\|F(v^*)\|)\exp(L_1 C(\omega)) + \|F(v^*)\|$.

Now, for both cases $\alpha \in (0, 1)$ and $\alpha = 1$ in (116) and (119), respectively, we have that $\|F(h_k(\omega))\|$ is upper bounded by $\max\{C_1(\omega), \overline{C}_1(\omega)\}$. Using this and a comparison test, we obtain

$$\begin{aligned}\sum_{k=0}^\infty \beta_k \min\left\{1, \frac{1}{\|F(h_k(\omega))\| + 2\sigma}\right\} &\geq \sum_{k=0}^\infty \beta_k \min\left\{1, \frac{1}{\max\{C_1(\omega), \overline{C}_1(\omega)\} + 2\sigma}\right\} \\ &= \min\left\{1, \frac{1}{\max\{C_1(\omega), \overline{C}_1(\omega)\} + 2\sigma}\right\}\sum_{k=0}^\infty \beta_k \quad (120) \\ &= \infty,\end{aligned}$$

where the last equality holds by $\sum_{i=0}^\infty \beta_k = \infty$. Thus,

$$\sum_{k=0}^\infty \mathbb{E}[\gamma_k \mid \mathcal{F}_{k-1}] = \infty \quad a.s. \quad (121)$$

Since, $\sum_{k=0}^\infty \mathbb{E}[\gamma_k \mid \mathcal{F}_{k-1}] = \infty$ almost surely, from (107) it follows that

$$\liminf_{k \to \infty} \mathrm{dist}^p(u_k, U^*) = 0 \quad a.s. \quad (122)$$

By Lemma 4.1, the sequence $\{\|h_k - u^*\|\}$ converges *a.s.* for any given $u^* \in U^*$. Thus, the sequence $\{h_k\}$ is bounded *a.s.* and, consequently, it has accumulation points *a.s.* In view of relation (20) in Lemma 4.1, it follows that

$$\lim_{k \to \infty} \|h_k - u_k\| = 0 \quad a.s. \quad (123)$$

Therefore, the sequences $\{u_k\}$ and $\{h_k\}$ have the same accumulation points a.s.

Now, let $\{k_i \mid i \geq 1\}$ be a (random) index sequence such that

$$\lim_{i \to \infty} \operatorname{dist}^p(u_{k_i}, U^*) = \liminf_{k \to \infty} \operatorname{dist}^p(u_k, U^*) = 0 \quad a.s. \tag{124}$$

Without loss of generality, we may assume that $\{h_{k_i}\}$ is a convergent sequence (for otherwise, we will select a convergent subsequence), and let $\bar{u}$ be its (random) limit point, i.e.,

$$\lim_{i \to \infty} \|h_{k_i} - \bar{u}\| = 0 \qquad a.s. \tag{125}$$

By relation (20), it follows that $\lim_{k \to \infty} \|h_k - u_k\| = 0$ a.s., which in view of the preceding relation implies that

$$\lim_{i \to \infty} \|u_{k_i} - \bar{u}\| = 0 \qquad a.s.$$

By continuity of the distance function $\operatorname{dist}(\cdot, U^*)$, from relation (124) we conclude that $\operatorname{dist}(\bar{u}, U^*) = 0$ a.s., which implies that $\bar{u} \in U^*$ almost surely since the set $U^*$ is closed. Since the sequence $\{\|h_k - u^*\|^2\}$ converges a.s. for any $u^* \in U^*$, it follows that $\lim_{k \to \infty} \|h_k - \bar{u}\| = 0$ a.s. By relation (123) we conclude that $\lim_{k \to \infty} \|u_k - \bar{u}\| = 0$ a.s. $\blacksquare$

### C.3 PROOF OF LEMMA 4.3

*Proof.* The choice of parameters $\beta_k$, ensures that $1 - 6\beta_k^2(K_0 + K_1 + K_3)^2 \geq 1/2$. Then, by taking the expectation in (19) of Lemma 4.1 and using Assumption 2.5, and definition of $C_e(\beta_k, \alpha) = K_1$ for $\alpha \in (0, 1)$, we obtain

$$\begin{aligned} \mathbb{E}[\|h_{k+1} - u^*\|^2] &\leq \mathbb{E}[\|h_k - u^*\|^2] - \frac{1}{2}\mathbb{E}[\|h_k - u_k\|^2] - 2\mathbb{E}[\gamma_k \mu \operatorname{dist}^p(u_k, U^*)] \\ &\quad + 6\beta_k^2(\sigma^2 + K_1\sigma^{2\alpha}). \end{aligned} \tag{126}$$

The equation (126) satisfies the condition of Lemma A.3 with

$$\bar{v}_k = \mathbb{E}[\|u_k - u^*\|^2], \quad \bar{a}_k = 0, \quad \bar{b}_k = 6\beta_k^2(\sigma^2 + K_1\sigma^{2\alpha}),$$

$$\bar{z}_k = 2\mu\mathbb{E}[\gamma_k \operatorname{dist}^p(u_k, U^*)] + \frac{1}{2}\mathbb{E}[\|h_k - u_k\|^2]. \tag{127}$$

By Lemma A.2, it follows that the sequence $\mathbb{E}[\|h_k - u^*\|^2]$ converges to a non-negative scalar for any $u^* \in U^*$. Since the sequence $\{\mathbb{E}[\|h_k - u^*\|^2]\}$ converges for all $u^* \in U^*$, it follows that the sequence $\{\mathbb{E}[\|h_k - u^*\|^2]\}$ is bounded for all $u^* \in U^*$. Next, using property of $\alpha$-symmetric operators, we show that $\mathbb{E}[\|F(h_k)\|]$ is bounded for all $k \geq 0$. Let $v^* \in U^*$ be an arbitrary but fixed solution. Since $\alpha \leq 1/2$, it holds that

$$\begin{aligned} \|F(h_k)\| &\leq \|F(h_k) - F(v^*)\| + \|F(v^*)\| \\ &\leq \|h_k - v^*\|(K_0 + K_1\|F(v^*)\|^\alpha + K_2\|h_k - v^*\|^{\alpha/(1-\alpha)}) + \|F(v^*)\|. \end{aligned} \tag{128}$$

Taking the expectation, we obtain

$$\mathbb{E}[\|F(h_k)\|] \leq (K_0 + K_1\|F(v^*)\|^\alpha)\mathbb{E}[\|h_k - v^*\|] + K_2\mathbb{E}[\|h_k - v^*\|^{1+\alpha/(1-\alpha)}] + \|F(v^*)\|. \tag{129}$$

Notice that $\mathbb{E}[\|h_k - v^*\|^{1+\alpha/(1-\alpha)}] = \mathbb{E}[(\|h_k - v^*\|^2)^{1/2(1-\alpha)}]$ and, for $\alpha \leq 1/2$, the quantity $1/2(1-\alpha) \leq 1$. Thus, we can apply Jensen inequality for concave function

$$\mathbb{E}[(\|h_k - v^*\|^2)^{1/2(1-\alpha)}] \leq \mathbb{E}[\|h_k - v^*\|^2]^{1/2(1-\alpha)}.$$

Therefore, using the preceding relation and Jensen inequality for the first term on the RHS of equation (129), we obtain

$$\mathbb{E}[\|F(h_k)\|] \leq (K_0 + K_1\|F(v^*)\|^\alpha)\mathbb{E}[\|h_k - v^*\|^2]^{1/2} + K_2\mathbb{E}[\|h_k - v^*\|^2]^{1/2(1-\alpha)} + \|F(v^*)\|. \tag{130}$$

Since $\mathbb{E}[\|h_k - v^*\|^2]$ is bounded, it follows that $\mathbb{E}[\|F(h_k)\|]$ is bounded by some constant $C_F > 0$ for all $k \geq 0$. $\blacksquare$

## C.4 PROOF OF THEOREM 4.4

*Proof.* The choice of the parameters $\beta_k$ ensures that $1 - 6\beta_k^2(K_0 + K_1 + K_2)^2 \geq \frac{1}{2}$, then by letting $u^* = P_{U^*}(h_k)$ in (104) in the proof of Lemma 4.1, with $C_e(\beta_k, \alpha) = K_1$, we get

$$\|h_{k+1} - P_{U^*}(h_k)\|^2 \leq \text{dist}^2(h_k, U^*) - \frac{1}{2}\|h_k - u_k\|^2 - 2\gamma_k \mu \text{dist}^p(u_k, U^*)$$
$$- 2\gamma_k \langle e_k^2, u_k - u^* \rangle + 3\beta_k^2(\|e_k^2\|^2 + \|e_k^1\|^2 + 2K_1\|e_k^1\|^{2\alpha}). \tag{131}$$

By the definition of the distance function, we have

$$\text{dist}^2(u_{k+1}, U^*) \leq \|u_{k+1} - P_{U^*}(u_k)\|^2.$$

Thus,

$$\text{dist}^2(h_{k+1}, U^*) \leq \text{dist}^2(h_k, U^*) - \frac{1}{2}\|h_k - u_k\|^2 - 2\gamma_k \mu \text{dist}^p(u_k, U^*)$$
$$- 2\gamma_k \langle e_k^2, u_k - u^* \rangle + 3\beta_k^2(\|e_k^2\|^2 + \|e_k^1\|^2 + 2K_1\|e_k^1\|^{2\alpha}). \tag{132}$$

Next, we estimate the term $\text{dist}^p(u_k, U^*)$ in (132). By the triangle inequality, we have

$$\|h_k - u^*\| \leq \|u_k - h_k\| + \|u_k - u^*\| \qquad \text{for all } u^* \in U^*,$$

and by taking the minimum over $u^* \in U^*$ on both sides of the preceding relation, we obtain

$$\text{dist}(h_k, U^*) \leq \|u_k - h_k\| + \text{dist}(u_k, U^*). \tag{133}$$

Applying Lemma A.5 with $p > 0$ in equation (133) yields

$$\text{dist}^p(h_k, U^*) \leq (\|u_k - h_k\| + \text{dist}(u_k, U^*))^p$$
$$\leq 2^{p-1}\|u_k - h_k\|^p + 2^{p-1}\text{dist}^p(u_k, U^*). \tag{134}$$

Using projection inequality (26), and stepsizes choice (16), we obtain

$$\|u_k - h_k\| \leq \|\gamma_k \Phi(h_k, \xi_k^1)\| \leq 1.$$

Combining this result with equation (134), with $p \geq 2$, we get

$$\text{dist}^p(h_k, U^*) \leq 2^{p-1}\|u_k - h_k\|^{2+(p-2)} + 2^{p-1}\text{dist}^p(u_k, U^*)$$
$$\leq 2^{p-1}\|u_k - h_k\|^2 + 2^{p-1}\text{dist}^p(u_k, U^*). \tag{135}$$

By dividing the relation in (135) with $2^{p-1}$ and by rearranging the terms, we obtain the following relation

$$-\text{dist}^p(u_k, U^*) \leq \|u_k - h_k\|^2 - 2^{1-p}\text{dist}^p(h_k, U^*). \tag{136}$$

Combining the preceding inequality with (132), we find that for any $k \geq 0$,

$$\text{dist}^2(h_{k+1}, U^*) \leq \text{dist}^2(h_k, U^*) - 2^{2-p}\mu\gamma_k\text{dist}^p(h_k, U^*) - \frac{1}{2}\|u_k - h_k\|^2 + 2\mu\gamma_k\|u_k - h_k\|^2$$
$$- 2\gamma_k \langle e_k^2, u_k - u^* \rangle + 3\beta_k^2(\|e_k^2\|^2 + \|e_k^1\|^2 + 2K_1\|e_k^1\|^{2\alpha}). \tag{137}$$

By the choice of $\beta_k$, we have $\beta_k = \dfrac{2}{a(\frac{2d}{a} + k)}$, where $a = \mu \min\left\{1, \frac{1}{2(C_F+\sigma)}\right\}$ and $d \geq 4\mu$. Thus, for all $k \geq 0$,

$$\beta_k \leq \frac{1}{d} \leq \frac{1}{4\mu} \qquad \implies \qquad 2\mu\beta_k \leq \frac{1}{2}.$$

By the definition of the stepsize $\gamma_k$, we always have $\gamma_k \leq \beta_k$. Therefore, $2\mu\gamma_k \leq 2\mu\beta_k \leq \frac{1}{2}$ for all $k \geq 0$, thus implying that

$$-\frac{1}{2}\|u_k - h_k\|^2 + 2\mu\gamma_k\|u_k - h_k\|^2 \leq 0. \tag{138}$$

Using the stochastic properties of $\xi_k$ imposed by Assumption 2.5, we have for all $k \geq 0$,

$$\mathbb{E}[\mathbb{E}[\gamma_k \mathbb{E}[\langle e_k^2, u_k - u^* \rangle \mid \mathcal{F}_{k-1} \cup \{\xi_k^1\}] \mid \mathcal{F}_{k-1}]] = 0,$$

$$\mathbb{E}[\mathbb{E}[\|e_k^2\|^2 \mid \mathcal{F}_{k-1} \cup \{\xi_k^1\}]] \leq \sigma^2, \qquad \mathbb{E}[\mathbb{E}[\|e_k^1\|^2 \mid \mathcal{F}_{k-1}]] \leq \sigma^2. \tag{139}$$

Moreover, since $\alpha \leq 1$ then the conditional expectation $\mathbb{E}[\|e_k^1\|^{2\alpha} \mid \mathcal{F}_{k-1}]$ is defined, and by Jensen inequality $\mathbb{E}[\|e_k^1\|^{2\alpha} \mid \mathcal{F}_{k-1}] \leq \sigma^{2\alpha}$ for all $k \geq 0$. Thus, by taking the total expectation in relation (137) and using an estimate from (138), we obtain for all $u^* \in U^*$ and for all $k \geq 0$,

$$\mathbb{E}[\text{dist}^2(h_{k+1}, U^*)] \leq \mathbb{E}[\text{dist}^2(h_k, U^*)] - 2^{2-p}\mu\mathbb{E}[\gamma_k \text{dist}^p(h_k, U^*)] + 6\beta_k^2(\sigma^2 + K_1\sigma^{2\alpha}). \tag{140}$$

The equation (140) is similar to equation (70) in the proof of Theorem 3.4, with the same stepsize structure. Thus, by following the same arguments from equations (70) to equation (76) in the proof of Theorem 3.4, we arrive at

$$\mathbb{E}[\text{dist}^2(h_{k+1}, U^*)] \leq \mathbb{E}[\text{dist}^2(h_k, U^*)] - 2^{2-p}\mu\beta_k \min\left\{1, \frac{1}{2(C_F + \sigma)}\right\} \mathbb{E}[\text{dist}^p(h_k, U^*)]$$

$$+ 6\beta_k^2(\sigma^2 + K_1\sigma^{2\alpha}), \tag{141}$$

where $C_F$ is an upperbound on $\mathbb{E}[\|F(h_k)\|]$ from the statement of Lemma 4.3. Now let $D_k = \mathbb{E}[\text{dist}^2(h_k, U^*)]$, and consider two cases $p = 2$ and $p > 2$.

**Case $p = 2$.** We note that by the definition of $a = \mu\min\left\{1, \frac{1}{2(C_F+\sigma)}\right\}$ and $d$, we have that $d \geq 4\mu$ and $\mu \geq a$, implying that $d \geq a$. Hence, for $p = 2$, relation (141) satisfies the conditions of Lemma A.6 with the following identification

$$r_k = D_k, \quad a = \mu\min\left\{1, \frac{1}{2(C_F + \sigma)}\right\}, \quad \alpha_k = \beta_k, \quad s_k = 0, \quad c = 6(\sigma^2 + K_1\sigma^{2\alpha}). \tag{142}$$

Therefore, for the choice $\beta_k = \frac{2}{a(\frac{2d}{a} + k)}$, we get the following convergence rate for all $k \geq 1$,

$$D_{k+1} \leq \frac{8d^2 D_0}{a^2 k^2} + \frac{12(\sigma^2 + K_1\sigma^{2\alpha})}{a^2 k}. \tag{143}$$

**Case $p > 2$.** From (141) by using $D_k = \mathbb{E}[\text{dist}^2(h_k, U^*)]$ and $a = \mu\min\left\{1, \frac{1}{2(C_F+\sigma)}\right\}$, we obtain for all $k \geq 0$,

$$D_{k+1} \leq D_k - 2^{2-p}a\beta_k\mathbb{E}[\text{dist}^p(h_k, U^*)] + 6\beta_k^2(\sigma^2 + K_1\sigma^{2\alpha}). \tag{144}$$

When $p > 2$, by applying Jensen inequality, we obtain

$$\mathbb{E}[\text{dist}^p(h_k, U^*)] = \mathbb{E}\left[\left(\text{dist}^2(h_k, U^*)\right)^{p/2}\right] \geq \left(\mathbb{E}[\text{dist}^2(h_k, U^*)]\right)^{p/2} = (D_k)^{p/2}.$$

Thus, for all $k \geq 0$, we have

$$D_{k+1} \leq D_k - 2^{2-p}a\beta_k(D_k)^{p/2} + 6(\sigma^2 + K_1\sigma^{2\alpha})\beta_k^2. \tag{145}$$

By telescoping the inequalities in (145), we obtain for $k \geq 0$,

$$D_{k+1} \leq D_0 - 2^{2-p}a\sum_{t=0}^{k}\beta_t(D_t)^{p/2} + 6(\sigma^2 + K_1\sigma^{2\alpha})\sum_{t=0}^{k}\beta_t^2. \tag{146}$$

Let $D_k^{\text{best}} = \min_{t=[0,...,k]} D_t$, then, by rearranging the term we get for all $k \geq 0$,

$$(D_k^{\text{best}})^{p/2}\sum_{t=0}^{k}\beta_t \leq \sum_{t=0}^{k}\beta_t(D_t)^{p/2} \leq \frac{D_0 - D_{k+1} + 6(\sigma^2 + K_1\sigma^{2\alpha})\sum_{t=0}^{k}\beta_t^2}{2^{2-p}a}. \tag{147}$$

Now, we use the choice for $\beta_k$, i.e., $\beta_k = \frac{b}{(k+1)^q}$, where $0 < b < \frac{1}{2\sqrt{3}(K_0+K_1+K_2)}$ and $1/2 < q < 1$. Then, the sequence $\{\beta_k\}$ satisfies the conditions of Lemma 4.3. Furthermore, by Lemma A.7, we have that for all $k \geq 1$,

$$\sum_{t=0}^{k} \beta_t \geq \frac{b}{1-q}((k+1)^{1-q} - 2^{1-q}), \qquad \sum_{t=0}^{k} \beta_t^2 \leq \frac{b^2}{2q-1}. \tag{148}$$

Combining equations (147) and (148), and omitting $D_{k+1}$, we obtain for all $k \geq 1$,

$$(D_k^{\text{best}})^{p/2} \leq \frac{2^{p-2}(1-q)\left(D_0 + 6b^2(\sigma^2 + K_1\sigma^{2\alpha})(2\sigma^2+1)/(2q-1)\right)}{ab\left((k+1)^{1-q} - 2^{1-q}\right)}. \tag{149}$$

Raising both sides of the preceding inequality in power $2/p$, we have that for all $k \geq 1$,

$$D_k^{\text{best}} \leq \frac{2^{2(p-2)/p}(1-q)^{2/p}\left(D_0 + 6b^2(\sigma^2 + K_1\sigma^{2\alpha})(2\sigma^2+1)/(2q-1)\right))^{2/p}}{(ab)^{2/p}\left((k+1)^{1-q} - 2^{1-q}\right)^{2/p}}. \tag{150}$$

$\blacksquare$

## D  ADDITIONAL EXPERIMENTS

We investigate the robustness of the methods for a larger choice of the initial parameter value $\beta_0$. In Figure 3, we set $q = 1 - \epsilon$, and corresponding $\beta_k = \frac{50}{10+k^{1-\epsilon}}$, so the initial stepsize $\beta_0 \approx 5$ and run all four methods for the same problem parameter choice. We observe that for a generalized smooth SVI when we increase stepsizes, the performance of clipped stochastic Popov and Korpelevich is comparable to that of both clipped stochastic versions. While in smooth SVI, the stepsizes for stochastic Korplevich and Popov methods can be much larger than for stochastic projection methods, improving the convergence performance of stochastic Korplevich and Popov methods.

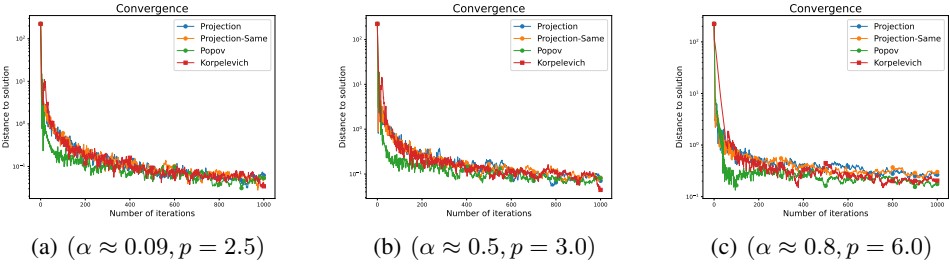

(a) ($\alpha \approx 0.09, p = 2.5$)     (b) ($\alpha \approx 0.5, p = 3.0$)     (c) ($\alpha \approx 0.8, p = 6.0$)

Figure 3: Comparison of the clipped stochastic projection, same-sample projection, Korpelevich, and Popov methods with $\beta = 50/(10 + k^{1-\epsilon})$.

