# OpenReview forum: "Generalized Smooth Stochastic Variational Inequalities:  Almost Sure Convergence and Convergence Rates"
_ICLR.cc/2025/Conference — Submitted to ICLR 2025_

### Official Review · Reviewer_KJ6o · 2024-10-31

**Soundness:** 3
**Presentation:** 3
**Contribution:** 3
**Rating:** 5
**Confidence:** 4

**Summary:**

The paper focuses on solving Stochastic Variational Inequalities (SVI) under a generalized smoothness assumption. More specifically, the authors analyze two methods, the Projection method and the Korpelevich method, in the quasi-sharp regime. The asymptotic convergence of both algorithms is established under the general smoothness assumption. In addition, the authors provide theoretical guarantees for the last iterate and establish the rate of convergence of both methods. Experimental results compare the two methods, validating the theoretical results for different stepsizes.

**Strengths:**

- The convergence is established under an assumption that is more general than smoothness.
- The paper establishes asymptotic convergence for two algorithms in the quasi-sharp setting.
- The associated rates of convergence are provided for each algorithm.

**Weaknesses:**

- Even though the paper provides a good amount of previous work on the topic, the positioning of the new results in the established literature is not done sufficiently. It would be nice to have a comparison of the provided convergence rates for the two methods with existing results. For example, one can compare the rate of the Projection method with the algorithm in [1] in the special setting of minimax problems.
- Based on the start of Section 3, it seems that two stochastic oracles are needed in the Projection method for the analysis to go through. This approach seems to require double the number of stochastic oracles than the classical Stochastic Projected Gradient Descent Ascent (or Stochastic Projection method with one oracle per iteration), which is not favorable.
- It would be nice to have a comparison between the two methods, Projection algorithm and Korpelevich method. Even though there is a reference to that in the Experimental section, a precise comparison of the two algorithms and their rate after providing the rates of convergence would be beneficial to have.
- The assumption on bounded variance of the stochastic oracles might be strong in some settings.

Some Typos:
- In the introduction (line 57): “normalizedChen et al. (2023b)” the reference is not correctly inserted.
- In section 1.1: “SVIs Jelassi et al. (2022)” the reference is not correctly inserted.
- In section 1.1: “Generalized Smooth Optimization In Zhang et al. (2019)” the reference needs to be correctly inserted.
- In section 3.2, it is mentioned “Then, from Theorem 4.4 we obtain”. However, I think that Theorem 3.4 was meant to be mentioned.


References

[1]: Delving into the Convergence of Generalized Smooth Minimax Optimization, W Xian, Z Chen, H Huang, ICML 2024

**Questions:**

- Since the analysis of the stochastic projection method seems to require two stochastic oracles, one for the update step and one for the computation of the stepsize, the total number of stochastic oracles needed would be twice the number of oracles the same method would require when analyzed without the generalized smoothness assumption. Do you think that this is a limitation of the current proof technique or it is required in general for establishing convergence under the more relaxed smoothness condition?
- Which of the two methods, stochastic Projection or Stochastic Korpelevich, should be preferred in the quasi-sharp regime?

**Details Of Ethics Concerns:**

No ethics concerns.

---

> ### Author Response · Authors · 2024-11-20
> **Response to Review by Reviewer KJ6o**
>
> We sincerely thank the reviewer for their valuable feedback and constructive recommendations, which have helped us improve the quality and clarity of our paper. We also corrected all noted typos.
> - **Rate Comparison with Related Works**: We have added detailed discussions comparing our rates with related works after Theorems 3.4 and 4.4 (see lines 329–334 and 438–443). Additionally, we included a recommended discussion on Xian et al. 2024 [1] in lines 84–89. Observe that Xian et al. 2024 [1] considered generalized smooth stochastic nonconvex strongly-concave min-max problems and provided $\mathcal{O}(\frac{1}{k^{1/4}})$ convergence rates for variants of stochastic gradient sescent ascent (SGDA) with normalized and adaptive stepsizes. The specific structure of the minmax problem and the fact that the gradient of the corresponding function is nonmonotone in one variable and strongly monotone in another makes it difficult to directly apply their results to SVIs. While both settings are interesting, direct comparisons of results from Xian et al 2024 and ours are challenging due to these differences. One possible example that satisfies the assumption from Xian et al. 2024 [1] and our assumptions is a generalized smooth, strongly convex strongly concave min-max problem. For this problem, the rate of SGDA from Corollary 4.16 is $\mathcal{O}(k^{-1/4})$, while ours is $\mathcal{O}(k^{-1})$ (since $p=2$) which is much better. However, we acknowledge that the differences in assumptions and problem structures limit this comparison. We might consider including this example problem in our manuscript, but we believe the fundamental distinctions between the works should be considered.
>
> - **Two samples:** Theorems 3.1, 3.2 in Koloskova et al. 2023 [2] suggest that there exists an operator (which does satisfy our assumptions) for which there exists a fixed point of standard clipping SGD, which is not a solution, which leads to unavoidable bias for one sample clipped SGD. Using two samples helps us to overcome unavoidable bias issues present in one-sample clipped SGD, as shown in Koloskova et al. 2023 [2], and ensures both almost-sure convergence and unbiased convergence rates. Please see lines 280–290 in the revised manuscript for further clarification. Furthermore, utilization of two samples per iteration does not affect dependency on the number of iterations $k$. However, developing a method that requires only one sample per iteration would be an interesting further direction of research.
> - **Comparison of Projection and Korpelevich Methods:** We have included a comparison of the rate for stochastic clipped projection and Korpelevich methods after Theorem 4.4. Both methods exhibit the same dependency on $k$ for convergence rates, in cases where $p=2$ and $p\geq 2$.
>
> - **Bounded Variance Assumption:** We agree with the reviewer that the bounded variance assumption is strong and could potentially be relaxed. However, even under this assumption, the analysis in the generalized smooth setting presents significant challenges. To achieve convergence even under bounded variance assumption, one needs to use or develope  new techniques such as bounding gradients over trajectories (Li et al., 2023 [3]) or demonstrating the almost sure non-summability of the series of step sizes, as in our work. It is worth noting that even in the original work by Chen et al. 2023 [4], the authors adopted a stronger assumption than bounded variance. Relaxing this assumption remains an interesting direction for future research.
>
> - **Comparison in Theory and Experiments:** Based on both theoretical results and experimental findings, the stochastic clipped projection and Korpelevich methods are comparable under the assumptions of $p$-quasi sharpness and $\alpha$-symmetry.
>
> [1] Delving into the convergence of generalized smooth minimax optimization, Wenhan Xian, Ziyi Chen, and Heng Huang. , ICML 2024.
>
> [2] Revisiting gradient clipping: Stochastic bias and tight convergence guarantees, Anastasia Koloskova, Hadrien Hendrikx, and Sebastian U Stich, ICML 2023
>
> [3] Convex and non-convex optimization under generalized smoothness. Haochuan Li, Jian Qian, Yi Tian, Alexander Rakhlin, and Ali Jadbabaie, NeurIPS 2023
>
> [4] Generalized-smooth nonconvex optimization is as efficient as smooth nonconvex optimization, Ziyi Chen, Yi Zhou, Yingbin Liang, and Zhaosong Lu, ICML, 2023

---

> > ### Comment · Reviewer_KJ6o · 2024-11-23
> >
> > Thank you very much for your response.
> > I will stick to the current score.

---

### Official Review · Reviewer_HPK3 · 2024-11-01

**Soundness:** 3
**Presentation:** 3
**Contribution:** 2
**Rating:** 6
**Confidence:** 3

**Summary:**

The paper considers $L_0,L_1$-smooth stochastic variational inequalities satisfying a generalization of strong monotonicity/coherence called $p$-quasi sharpness.
For both a clipped variant of projection stochastic gradient method and the projected stochastic extragradient method they show:

- $\mathcal O(1/k)$ rates in expectation for the last iterate when $p=2$ (strongly coherent problems), matching the one under (standard) Lipschitz continuity (e.g. Hsieh et al. 2019b)
- $\mathcal O(k^{-2(1-q)/p})$ rates for the best iterate for $q\in (1,1/2)$.
- With a Robbins-Monro stepsize factor they show almost sure convergence.

Both methods uses two gradient computations since the gradient method requires the clipping to be performed with a fresh sample.

**Strengths:**

- The paper reads very easily and is transparent about contributions
- Relaxing smoothness assumptions for VIs seems like an interesting direction

**Weaknesses:**

- Assuming $p$-quasi sharpness seems like a strong assumption since even (clipped) gradient descent suffice in this setting, making it close to the minimization case. In light of this the result does not seem particularly surprising.
- considering that gradient descent suffice I do not understand the motivation for considering the extragradient variant. It would be helpful if the authors could elaborate on the benefit/reason. I could see it being interesting if clipped extragradient also have a guarantees for bilinear or convex-concave problems (in which case the guarantee would only be for the average in light of e.g. Hsieh et al. 2020), but this is not currently provided.
- Even the clipped gradient method requires 2 gradient computations, which seems unconventional (cf. e.g. Gorbunov et al. 2022). It would be interesting if the additional gradient call could be removed.

Minor:

- l. 481 maybe don't state that p-quasi-sharp is a wide class, consider the (weak) Minty variational inequality also mentioned later.
- l. 489 $\mu < 0$ does not seems to correspond to the weak Minty variational inequality (you need the operator evaluation on the right hand side instead of solution distance). It might be worth also providing a reference.

Typos:

- l. 99 stepszies -> stepsizes
- l. 105 should be $O(k^{-1})$
- l. 75 thr -> the

**Questions:**

- Eq. 9: could you elaborate why you need a fresh sample? Is this an artifact of the analysis?
- It is curious that Popov's method dominates all other methods in the experiments:
     - Is there any intuition for why this is the case?
    - It is not clear to me how the clipping is performed for Popov since each update involves to gradients (the old and new). Is the old gradient clipped with a fresh sample, such that the number of gradient calls are the same as for the clipped version of the gradient/extragradient method?
     - What is the difficulty in analysing a clipped version of Popov's method under generalization smoothness?
- It seems strange that a large choice of $q$ improves performance, while the theory suggests otherwise. What are possible explanations?
- As a sanity check do you recover Theorem 2.5 of Koloskova et al. (2023) in the strongly convex deterministic case?

---

> ### Author Response · Authors · 2024-11-20
>
> We sincerely thank the reviewer for their valuable feedback and constructive comments. We are pleased that the reviewer found our work correct, transparent, and an important direction for relaxing smoothness assumptions in variational inequalities. Next, we address the concerns raised by the reviewer and respond to their questions:
> - **p-Quasi Sharpness**: We respectfully disagree with the reviewer’s assessment that the derived results are unsurprising. The main difficulty of our analysis lies in the generalized smoothness assumption, which makes the problem fundamentally different from smooth SVI. Even in the stochastic optimization case, one-sample clipped SGD fails to converge to a solution with diminishing step sizes (Koloskova et al., 2023 [1]). To the best of our knowledge, this is the first work to establish almost sure convergence and unbiased convergence rates for stochastic methods under generalized smoothness assumptions for SVIs. Additionally, SVIs differ significantly from optimization problems, as one cannot bound operator norms using function values (as it was done in Li et al., 2023 [2]; Koloskova et al., 2023 [1]; Xi, 2024 [3]). To address these challenges, we developed a novel technique demonstrating that a series of stochastic clipping step sizes is almost surely not summable.
> - **Motivation for Extragradient**: Both the projected gradient method and the extragradient method are widely used for SVI problems, and investigating their performance under generalized smoothness provides a comprehensive understanding of their behavior in more relaxed settings. While both methods exhibit the same convergence rate with respect to  $k$ , the extragradient method's convergence guarantees for bilinear or smooth convex-concave problems make it a valuable addition to this study of the generalized smooth SVIs. We agree with the reviewer that further exploration of extragradient methods under the Minty condition ($μ=0$) could be an interesting direction for future research.
> - **Two Samples for Gradient Clipping**: The use of two samples for gradient clipping, while unconventional, is  one of our key insights and contributions to our work.  We do agree with the reviewer that removing the second sample would be an interesting problem for further research. However, Theorems 3.1, 3.2 in Koloskova et al. 2023 suggest that there exists an operator (which does satisfy our assumptions) for which there exists a fixed point of standard clipping SGD, which is not a solution, which leads to unavoidable bias for one sample clipped SGD. This approach allows us to overcome unavoidable bias issues present in one-sample clipped SGD, as shown in Koloskova et al. (2023), and ensures both almost-sure convergence and unbiased convergence rates. Please see lines 280–290 in the revised manuscript for further clarification.
>
> - **Minor Issues**: Thank you for noticing these issues, we fixed them. We also corrected all noted typographical errors.
> - **Answers to Questions**:
> 	- We added the stochastic Popov method and its clipping step sizes in the Experiments section (lines 449–453). However, we are uncertain why stochastic clipped Popov performs better in the solution neighborhood.
> 	- The analysis of the stochastic clipped Popov method presents additional challenges due to the complexity of its clipping steps. We are open to including convergence results for this method if deemed beneficial.
> 	-  Based on our experiments, a large value of $q$ improves the performance of the methods in the $\sigma$-neighbourhood of the method. This can be explained by the fact that the stepsizes decrease faster for the large choice of  $q$. Thus, larger $q$ makes stepsizes smaller, leading methods to converge to a solution's smaller neighborhood.
> 	- Regarding Theorem 2.5 of Koloskova et al. (2023), we believe a direct comparison is not entirely appropriate due to differences in assumptions and settings. In our stochastic setting, decreasing stepsizes are necessary, while the stepsize in Theorem 2.5 in Koloskova et al [1] is fixed.  Also, it is well-known that while the rate of the projection (gradient) method for strongly monotone Lipshcitz VIs and strongly convex smooth optimization is linear, the factor inside $\exp$ is different: $\frac{L}{\mu}$ in optimization and $\frac{L^2}{\mu^2}$ in VIs. However, for $p=2$ when using fixed stepsizes in equation (76) of the proof of Theorem 3.4, it can be shown that the convergence rate to the neighborhood of a solution is also linear.
>
> [1] Revisiting gradient clipping: Stochastic bias and tight convergence guarantees, Anastasia Koloskova, Hadrien Hendrikx, and Sebastian U Stich, ICML 2023
>
> [2] Convex and non-convex optimization under generalized smoothness. Haochuan Li, Jian Qian, Yi Tian, Alexander Rakhlin, and Ali Jadbabaie, NeurIPS 2023
>
> [3] Delving into the convergence of generalized smooth minimax optimization, Wenhan Xian, Ziyi Chen, and Heng Huang. , ICML 2024.

---

> > ### Comment · Reviewer_HPK3 · 2024-11-25
> >
> > I thank the author's for clarifying. I only have a couple of remaining questions:
> >
> > **Two Samples for Gradient Clipping** it is stated that "two samples for gradient clipping [...] is one of our key insights and contributions to our work". Is this approach new even for minimization?
> >
> > **Motivation for Extragradient** The proposed EG method is still different from classical extragradient due to the stepsize choice. should one expect the proposed EG scheme to recover the rates for bilinear and smooth convex-concave problems?

---

> ### Author Response · Authors · 2024-11-25
>
> We thank the reviewer for their answer.
>
> 1. Yes, to the best of our knowledge, our two-sample approach is the first unbiased result of clipped SGD, even for optimization.
>
> 2. For smooth strongly convex strongly concave problems, which are a special case of $p=2$, we recover $O(\frac{1}{k})$ rate for stochastic clipped EG, which matches the rate for stochastic EG obtained in Kannan & Shanbhag, 2019 [1]. However, convex-concave problems do not satisfy the $p$-quasi sharpness assumption and are out of the scope of this work. We are not aware whether it is possible to show that the proposed variant of EG converges in the general setting of $\mu=0$, but it is an interesting open question for further research.
>
> [1] Aswin Kannan and Uday V Shanbhag. Optimal stochastic extragradient schemes for pseudomonotone stochastic variational inequality problems and their variants. Computational Optimization and Applications, 2019.

---

> > ### Comment · Reviewer_HPK3 · 2024-11-25
> >
> > I thank the author's for their transparent engagement and have decided to increase my score.

---

### Official Review · Reviewer_z6JX · 2024-11-03

**Soundness:** 3
**Presentation:** 2
**Contribution:** 2
**Rating:** 6
**Confidence:** 3

**Summary:**

The paper focuses on solving stochastic variational inequalities which do not satisfy the typical regularity (Lipschitz continuity) and structural assumptions (monotonicity). More precisely, the authors in order to extend the notion of Lipschitz continuity they consider the class of $\alpha-$ asymmetric operators (introduced by Chen 2023). Furthermore, their extension of non-monotonicity, they consider operators which satisfy the so-called $p$-quasi sharpness property. This structural assumption actually can be seen as a local type relaxation of strong convexity relative to the solutions.
To that end, the authors investigate under these assumption the behaviour of (stochastic) projected gradient descent and extra-gradient algorithm run with stochastic clipped step-sizes. In doing so, they provide asymptotic and non-asymptotic convergence guarantees for each respective method

**Strengths:**

The paper is clearly written and easy to follow. Furthermore the math supporting the main contribution of the paper, namely the theoretical convergence guarantees for two classical optimization algorithmic schemes run with stochastic clipped methods seems solid.
Moreover, as far as my knowledge goes the particular result are novel in the literature.
Finally, the numerical/experimental part seems clear and supports the theoretical results

**Weaknesses:**

Concerning the weaknesses, three observations are in order:

1.  The key innovation, compared to the variational coherent literature is the introduction of two independent samplings per iteration. In my opinion this needs further clarifications on how it affects the complexity of the rates both in theory and in the numerical experiments.

2. The under study relaxation of the Lipschitz continuity of the (VI) defining operator introduces some extra parameters which require an additional tuning effort (see the respective non-asymptotic convergence rate theorems). This fact requires an additional discussion concerning the practical efficiency of the methods.

3. The case of $p>2$ the respective guarantees are given with respect to the expected best distance.  To that end, this prohibits the proposal (due to the stochastic nature of the methods) of an actual output rule of the respective methods.

That being said, I still find the particular work interesting and look forward for a fruitful discussion period.

**Questions:**

See weaknesses section.

---

> ### Author Response · Authors · 2024-11-20
> **Response to Review by Reviewer z6JX**
>
> - We thank the reviewer for their valuable feedback and comments. We are delighted that the reviewer found our work important, novel, and correct.
>
> Below, we address the concerns raised by the reviewer and respond to their questions:
> - **Two Samples per Iteration**:  We have added a discussion in lines 329–332 addressing the use of two samples per iteration. While this approach requires twice as many oracle calls to achieve an $\epsilon$-solution, it does not affect the rate dependency on the number of iterations  $k$ . Using two samples per iteration allows us to achieve an unbiased rate, meaning that we can obtain an arbitrarily small  $\epsilon$ by decreasing the step sizes. In contrast, as shown in Koloskova et al. (2023), one-sample clipped SGD cannot achieve arbitrarily small  $\epsilon$ -solutions even with diminishing step sizes. This advantage highlights the trade-off between computational cost and achieving unbiased convergence guarantees, which we believe is an important contribution to the literature.
> - **Parameters**: We agree with the reviewer that the knowledge of the parameters  $L_0,L_1,K_0,K_1,K_2$  is crucial for our analysis. However, we note that the step sizes  $\beta_k$  for the clipped stochastic projection method in Theorem 3.4 (Case 2,  $p\geq2$ ) do not depend on problem-specific parameters. Furthermore, we believe that deriving optimal rates without prior knowledge of  $L_0$ ​ and ​ $L_1$  even in deterministic optimization—is an open problem. For reference, see Huber et al. (2024) [1].
> - **Best Iterate Convergence in Stochastic Cases**:  We acknowledge the reviewer’s concern that best-iterate convergence in stochastic cases is less tractable. To address this, we have modified the proof and demonstrated that the same rate can be achieved for the average-iterate convergence. Specifically, we show that the weighted point $\bar{u} = \frac{1}{\sum_{t=0}^k \beta_t} \sum_{t=0}^{k} \beta_t u_t$ sattisfies $\mathbb{E}[\rm{dist}^2( \bar{u}_k)] \leq \mathcal{O}(\frac{1}{k^{2(1-q)/p}})$ . This modification provides a more practical convergence guarantee. The result has been added to Theorems 3.4 and 4.4 in the revised manuscript.
>
> [1] Parameter-agnostic optimization under relaxed smoothness, Florian Hübler, Junchi Yang, Xiang Li, Niao He, AISTAT, 2024

---

> > ### Comment · Reviewer_z6JX · 2024-12-02
> > **Thank you for your reply**
> >
> > I thank the authors for their responses. My concerns are mainly answered. Given the discussion I am willing to keep my score unchanged.

---

> ### Author Response · Authors · 2024-12-01
>
> Dear Reviewer,
>
> We addressed all your concerns and improved our paper based on your recommendations. We kindly ask you to reconsider your evaluation of our paper.

---

### Official Review · Reviewer_4BxG · 2024-11-12

**Soundness:** 2
**Presentation:** 2
**Contribution:** 2
**Rating:** 3
**Confidence:** 3

**Summary:**

This paper focuses on stochastic variational inequality (SVI) problems involving non-monotone operators. Applying clipping to projected gradient descent and extragradient methods establishes almost-sure convergence and in-expectation convergence rates under generalized smoothness conditions, without boundedness assumptions.

**Strengths:**

This work relaxes several assumptions:
- uses a relaxed notion of L-Lipschitzness, called $\alpha$-symetric operator.
- does  not rely on the assumptions of bounded noise and bounded gradients

**Weaknesses:**

- Proposition 2.2. is written for an operator, and is taken from Proposition 1 in (Chen et al. 2023a). The latter is for a gradient of a real-valued function, the former is for an operator that is a more general vector field. It is unclear if Proposition 1 in (Chen et al. 2023a) directly extends to non-gradient field operators. The first equivalence (eq. 9) in  (Chen et al. 2023a) already fails for vector fields. Also, since the proof for (9) in  (Chen et al. 2023a) also relies on ODEs it involves solving the integral involving $h$ therein that contains a vector field. Could you please elaborate?


- Provide citations for the Extragradient and the Popov method.

- It is confusing that Chen et al 2023 is cited twice with the same title and authors.

# Writing

*Structure.* The contributions are listed in related works; perhaps introduce a paragraph.

*Abstract.*
- After reading the abstract I could not tell which setup this paper focuses on. It mentions "structured non-monotone operators" but it should be more specific.
- I could not tell which methods the paper focuses on. It states "well-known stochastic methods with clipping, namely, projection and Korpelevich". The former can be combined with all methods, so I was confused. For the latter, it is common to use 'extragradient'. Please use projected gradient descent (PGD) and projected extragradient throughout the paper. However, even with this change, the abstract is still confusing, as it gives the impression that you are focusing on problems beyond monotonicity, for which PGD last iterate does not converge; clarify if you are focusing on last, average, or best iterate.
After reading the abstract, I could not tell what rate you get and whether it matches the lower bound? Please add that.


*Other.*
- Use citing with brackets when needed
- line 49 missing citation for Adam
- line 57 missing citation for adaptive methods
- line 71 typo

**Questions:**

Please see above.

---

> ### Author Response · Authors · 2024-11-20
> **Response to Review by Reviewer 4BxG**
>
> - We thank the reviewer for their valuable feedback and comments. We have addressed all typos and missing or incorrect references, provided additional details in the abstract as recommended, and uploaded a revised version of our paper.
>
> - **Firstly**, we respectfully disagree with the reviewer’s concern regarding the correctness of Proposition 2.2. Proposition 1(a, b) in Chen et al. (2023) is general enough to apply to any vector field, not just gradient operators, as the proof does not use any information specific to gradient fields. Furthermore, Eq. (9) in Chen et al. (2023) is valid for  $\alpha$ -symmetric operators, as demonstrated in Lemma 2 (Appendix A) of their work, where $h(\theta) = L_0 + L_1 || F(\theta w' + (1 - \theta )w)||$ . To enhance clarity, we are willing to include additional clarifications or proofs of these results from Chen et al. (2023) in our manuscript, although we believe this might be redundant given their existing availability.
>
> - We have also corrected the citations for the Extragradient and Popov methods, removed the duplicate reference to Chen et al. (2023), and incorporated all other feedback regarding references and citations.
>
> - **Secondly**, we would like to highlight some contributions of our work that the reviewer might have overlooked. Our paper introduces a relaxation of the standard Lipschitz continuity assumption, presenting what we believe to be the first convergence results in the literature for generalized smooth stochastic variational inequalities. To the best of our knowledge, this work also provides the first almost-sure convergence results for stochastic clipping methods in a generalized smooth setting, even within optimization. Additionally, we introduced a novel analysis by demonstrating that the series of stochastic clipping step sizes is almost surely unsummable. For both methods considered, we established new unbiased convergence rates. These rates can't be achieved by the technique from Li et al 2024 or Xian et al 2024 since in SVIs one can not upperbound the gradient norm by the function residuals.
>
> - We hope that our responses and the revisions made to the manuscript address all of the reviewer’s concerns. We respectfully request the reviewer to reconsider their evaluation, as we believe the contributions of this work, particularly the novel analysis of generalized smoothness and clipped stochastic methods, significantly advance the field. We are looking forward to a discussion.

---

> ### Author Response · Authors · 2024-12-01
>
> Dear Reviewer,
>
> We have addressed all your concerns and improved our paper based on your recommendations. However, we believe the only technical weakness you provided is not valid, as Proposition 1 in (Chen et al., 2023a) applies to any vector field. We kindly ask the reviewer to reconsider their evaluation in light of these clarifications.

---

> > ### Comment · Reviewer_4BxG · 2024-12-02
> > **Thanks**
> >
> > I would have appreciated it if you addressed it in the revised version given that it is an important concern as you build on that result. It is not clear to me that it holds for a general vector field.

---

> ### Author Response · Authors · 2024-12-02
>
> Unfortunately, the last day that authors may upload a revised PDF was November 26, while we provided our response on November 19. While we cannot upload a new version at this stage, we will include these clarifications and results in the final version of the paper.
>
> > The first equivalence (eq. 9) in (Chen et al. 2023a) already fails for vector fields. Also, since the proof for (9) in (Chen et al. 2023a) also relies on ODEs it involves solving the integral involving
>  therein that contains a vector field. Could you please elaborate?
>
> The proof of (9) in Chen et al., 2023a, is provided in Appendix A, specifically in Lemma 2. To show that the $\alpha$-symmetric assumption implies (9), the authors relied on (i) the triangle inequality and (ii) the $\alpha$-symmetric assumption. These steps are valid for any vector field, which demonstrates that eq (9) **is valid** for any vector field.
>
> We kindly ask the reviewer to reconsider their evaluation in light of these clarifications.

---

### Meta-Review · Area_Chair_KhCs · 2024-12-19

**Metareview:**

The paper studies stochastic variational inequalities under generalized smoothness assumptions and for operators that are "quasi-sharp." Although the problem class is presented as corresponding to "structured non-monotone" operators, the quasi-sharpness is quite a strong assumption that excludes even some monotone problems (e.g., coming from bilinear problems) and suffices for GDA to converge (while GDA provably diverges for bilinear problems). The paper proves that GDA and EG methods with proposed step size & gradient estimation strategies (involving gradient clipping) exhibit almost sure convergence. They also provide complexity results for some of the considered settings, where the guarantees hold in expectation.

While the paper is technically solid, the motivation and the strength of contributions are not clear at this point. The paper would benefit from a better motivation for the considered problem class, better connection to and comparison to related work, and a broader discussion of how the results might extend to other cases of interest (at least handling all the monotone settings we know how to address at the moment).

**Additional Comments On Reviewer Discussion:**

The discussion centered around the points mentioned in the meta-review, concerning (1) relationship to prior work, (2) motivation, and (3) strength of contributions. The authors clarified that what they view as the major contribution is considering the generalized smoothness condition as opposed to handling non-monotone VIs. Yet, looking at the assumption, it is not clear it would cause major difficulties in the convergence analysis, especially given the "quasi-sharpness" assumption about the operator. Second, the motivation for considering the extragradient method given that GDA already works for the considered problem appears weak, even with the added clarification in the rebuttal. Finally, almost all the reviews raised the questions regarding the use of two samples per iteration. I think this last point was appropriately addressed by the authors in the rebuttal.

---

### Decision · Program_Chairs · 2025-01-22

Reject